# SwS: Self-aware Weakness-driven Problem Synthesis in Reinforcement Learning for LLM Reasoning

**Xiao Liang[1]\*, Zhong-Zhi Li[3]\*, Yeyun Gong[2]†, Yang Wang[2], Hengyuan Zhang[4],**
**Yelong Shen[2], Ying Nian Wu[1], Weizhu Chen[2]†**
[1] University of California, Los Angeles, [2] Microsoft
[3] School of Artificial Intelligence, Chinese Academy of Sciences, [4] Tsinghua University

## Abstract

Reinforcement Learning with Verifiable Rewards (RLVR) has proven effective for training large language models (LLMs) on complex reasoning tasks, such as mathematical problem solving. A prerequisite for the scalability of RLVR is a high-quality problem set with precise and verifiable answers. However, the scarcity of well-crafted human-labeled math problems and limited-verification answers in existing distillation-oriented synthetic datasets limit their effectiveness in RL. Additionally, most problem synthesis strategies indiscriminately expand the problem set without considering the model's capabilities, leading to low efficiency in generating useful questions. To mitigate this issue, we introduce a Self-aware Weakness-driven problem Synthesis framework (SwS) that systematically identifies model deficiencies and leverages them for problem augmentation. Specifically, we define weaknesses as questions that the model consistently fails to learn through its iterative sampling during RL training. We then extract the core concepts from these failure cases and synthesize new problems to strengthen the model's weak areas in subsequent augmented training, enabling it to focus on and gradually overcome its weaknesses. Without relying on external knowledge distillation, our framework enables robust generalization by empowering the model to self-identify and address its weaknesses in RL, yielding average performance gains of 10.0% and 7.7% on 7B and 32B models across eight mainstream reasoning benchmarks. Our code is available at https://github.com/MasterVito/SwS.

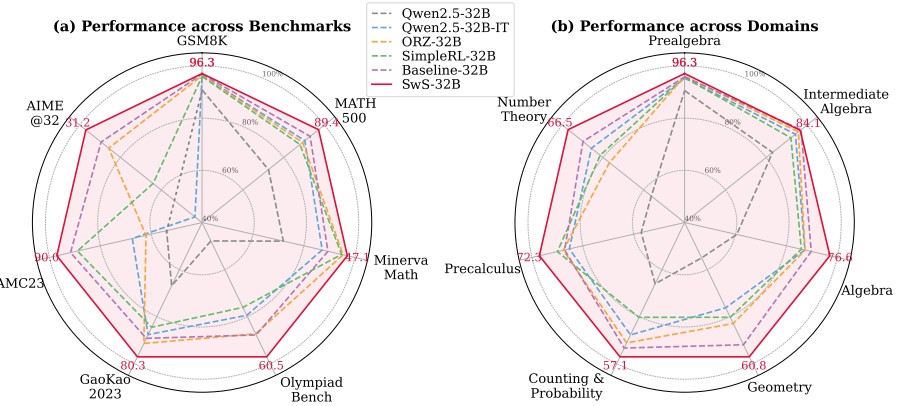

Figure 1: 32B model performance across mainstream reasoning benchmarks and different domains.

---

\*Equal contribution. Work done during Xiao's and Zhongzhi's internships at Microsoft.

†Corresponding authors: Yeyun Gong and Weizhu Chen (`yegong,wzchen@microsoft.com`)

39th Conference on Neural Information Processing Systems (NeurIPS 2025).

# 1  Introduction

*"Give me six hours to chop down a tree and I will spend the first four sharpening the axe."*

*—Abraham Lincoln*

Large-scale Reinforcement Learning with Verifiable Rewards (RLVR) has substantially advanced the reasoning capabilities of large language models (LLMs)[Jaech et al., 2024, Guo et al., 2025, Team et al., 2025], where simple rule-based rewards can effectively induce complex reasoning skills. The success of RLVR for eliciting models' reasoning capabilities heavily depends on a well-curated problem set with proper difficulty levels Yu et al. [2025b], Liu et al. [2025b], Xiong et al. [2025], where each problem is paired with an precise and verifiable reference answer [Hu et al., 2025, Luo et al., 2025, Yu et al., 2025b, Guo et al., 2025]. However, existing reasoning-focused datasets for RLVR suffer from three main issues: (1) High-quality, human-labeled mathematical problems are scarce, and collecting large-scale, well-annotated datasets with precise reference answers is cost-intensive. (2) Most reasoning-focused synthetic datasets are created for SFT distillation, where reference answers are rarely rigorously verified, making them suboptimal for RLVR, which relies heavily on the correctness of the final answer as the training signal. (3) Existing problem augmentation strategies typically involve rephrasing or generating variants of human-written questions [Yu et al., 2023, Luo et al., 2023, Pei et al., 2025, Liu et al., 2025a], or sampling concepts from existing datasets [Huang et al., 2024, Tang et al., 2024, Li et al., 2024a, Zhao et al., 2025b], without explicitly considering the model's reasoning capabilities. Consequently, the synthetic problems may be either too trivial or overly challenging, limiting their utility for model improvement in RL.

More specifically, in RL, it is essential to align the difficulty of training tasks with the model's current capabilities. When using group-level RL algorithms such as GPRO [Shao et al., 2024], the advantage of each response is calculated based on its comparison with other responses in the same group. If all responses are either entirely correct or entirely incorrect, the token-level advantages within each rollout collapse to 0, leading to gradient vanishing and degraded training efficiency Liu et al. [2025b], Yu et al. [2025b], and potentially harming model performance [Xiong et al., 2025]. Therefore, training on problems that the model has fully mastered or consistently fails to solve does not provide useful learning signals for improvement. However, a key advantage of the failure cases is that, unlike the overly simple questions with little opportunity for improvement, persistently failed problems reveal specific areas of weakness in the model and indicate directions for further enhancement. This raises the following research question: *How can we effectively utilize these consistently failed cases to address the model's reasoning deficiencies? Could they be systematically leveraged for data synthesis that targets the enhancement of the model's weakest capabilities?*

To answer these questions, we propose a Self-aware Weakness-driven Problem Synthesis (SwS) framework, which leverages the model's self-identified weaknesses in RL to generate synthetic problems for training augmentation. Specifically, we record problems that the model consistently struggles to solve or learns inefficiently through iterative sampling during a preliminary RL training phase. These failed problems, which reflect the model's weakest areas, are grouped by categories, leveraged to extract common concepts, and to synthesize new problems with difficulty levels tailored to the model's capabilities. To further improve weakness mitigation efficiency during training, the augmentation budget for each category is allocated based on the model's relative performance across them. Compared with existing problem synthesis strategies for LLM reasoning [Zhao et al., 2025b, Tang et al., 2024], our framework explicitly targets the model's capabilities and self-identified weaknesses, enabling more focused and efficient improvement in RL training.

To validate the effectiveness of SwS, we conducted experiments across model sizes ranging from 3B to 32B and comprehensively evaluated performance on eight popular mathematical reasoning benchmarks, showing that its weakness-driven augmentation strategy benefits models across all levels of reasoning capability. Notably, our models trained on the augmented problem set consistently surpass both the base models and those trained on the original dataset across all benchmarks, achieving a substantial average absolute improvement of 10.0% for the 7B model and 7.7% for the 32B model, even surpassing their counterparts trained on carefully curated human-labeled problem sets [Hu et al., 2025, Cui et al., 2025]. We also analyze the model's performance on previously failed problems and find that, after training on the augmented problem set, it is able to solve up to 20.0% more problems it had consistently failed in its weak domain when trained only on the original dataset. To further demonstrate the robustness and adaptability of the proposed SwS pipeline, we extend it to explore the

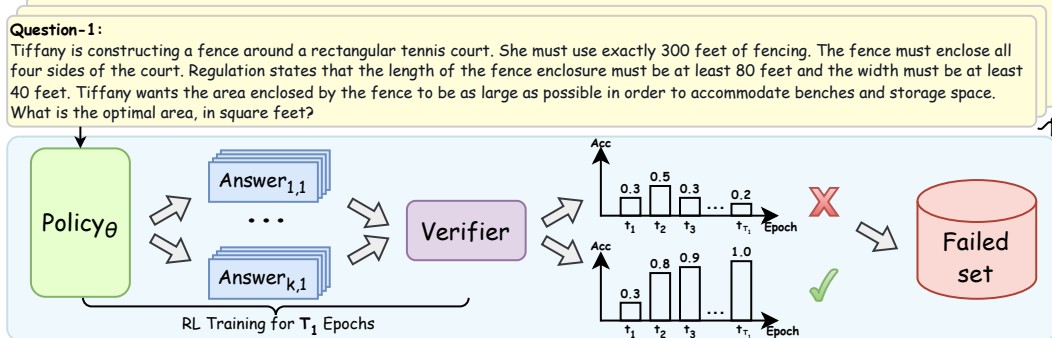

Figure 2: Illustration of the self-aware weakness identification during a preliminary RL training.

potential of *Weak-to-Strong Generalization*, *Self-evolving*, and *Weakness-driven Selection* settings, with detailed experimental results and analysis presented in Section 4.

**Contributions**. **(i)** We propose a Self-aware Weakness-driven Problem Synthesis (SwS) framework that utilizes the model's self-identified weaknesses to generate synthetic problems for enhanced RLVR training, paving the way for utilizing high-quality and targeted synthetic data for RL training. **(ii)** We comprehensively evaluate the SwS framework across diverse model sizes on eight mainstream reasoning benchmarks, demonstrating its effectiveness and generalizability. **(iii)** We explore the potential of extending our SwS framework to *Weak-to-Strong Generalization*, *Self-evolving*, and *Weakness-driven Selection* settings, highlighting its adaptability through detailed analysis.

## 2 Method

### 2.1 Preliminary

**Group Relative Policy Optimization (GRPO)**. GRPO Shao et al. [2024] is an efficient optimization algorithm tailored for RL in LLMs, where the advantages for each token are computed in a group-relative manner without requiring an additional critic model to estimate token values. Specifically, given an input prompt $x$, the policy model $\pi_{\theta_{\text{old}}}$ generates a group of $G$ responses $\mathbf{Y} = \{y_i\}_{i=1}^{G}$, with acquired rewards $\mathbf{R} = \{r_i\}_{i=1}^{G}$. The advantage $A_{i,t}$ for each token in response $y_i$ is computed as the normalized rewards:

$$A_{i,t} = \frac{r_i - \text{mean}(\{r_i\}_{i=1}^{G})}{\text{std}(\{r_i\}_{i=1}^{G})}. \tag{1}$$

To improve the stability of policy optimization, GRPO clips the probability ratio $k_{i,t}(\theta) = \frac{\pi_\theta(y_{i,t}|x,y_{i,<t})}{\pi_{\theta_{\text{old}}}(y_{i,t}|x,y_{i,<t})}$ within a trust region Schulman et al. [2017], and constrains the policy distribution from deviating too much from the reference model using a KL term. The optimization objective is defined as follows:

$$\mathcal{J}_{\text{GRPO}}(\theta) = \mathbb{E}_{x\sim\mathcal{D},\mathbf{Y}\sim\pi_{\theta_{\text{old}}}(\cdot|x)}$$

$$\left[\frac{1}{G}\sum_{i=1}^{G}\frac{1}{|y_i|}\sum_{t=1}^{|y_i|}\left(\min\left(k_{i,t}(\theta)A_{i,t},\ \text{clip}\left(k_{i,t}(\theta),1-\varepsilon,1+\varepsilon\right)A_{i,t}\right) - \beta D_{\text{KL}}(\pi_\theta||\pi_{\text{ref}})\right)\right]. \tag{2}$$

Inspired by DAPO Yu et al. [2025b], in all experiments of this work, we omit the KL term during optimization, while incorporating the *clip-higher*, *token-level loss* and *dynamic sampling* strategies to enhance the training efficiency of RLVR. Our RLVR training objective is defined as follows:

$$\mathcal{J}(\theta) = \mathbb{E}_{x\sim\mathcal{D},\mathbf{Y}\sim\pi_{\theta_{\text{old}}}(\cdot|x)}\left[\frac{1}{\sum_{i=1}^{G}|y_i|}\sum_{i=1}^{G}\sum_{t=1}^{|y_i|}\left(\min\left(k_{i,t}(\theta)A_{i,t},\ \text{clip}(k_{i,t}(\theta),1-\varepsilon,1+\varepsilon^h)A_{i,t}\right)\right)\right]$$

$$\text{s.t.}\ \ \text{acc}_{\text{lower}} < |\{y_i \in \mathbf{Y} \mid \text{is\_accurate}(x,y_i)\}| < \text{acc}_{\text{upper}}. \tag{3}$$

where $\varepsilon^h$ denotes the upper clipping threshold for importance sampling ratio $k_{i,t}(\theta)$, and $\text{acc}_{\text{lower}}$ and $\text{acc}_{\text{upper}}$ are thresholds used to filter target prompts for subsequent policy optimization.

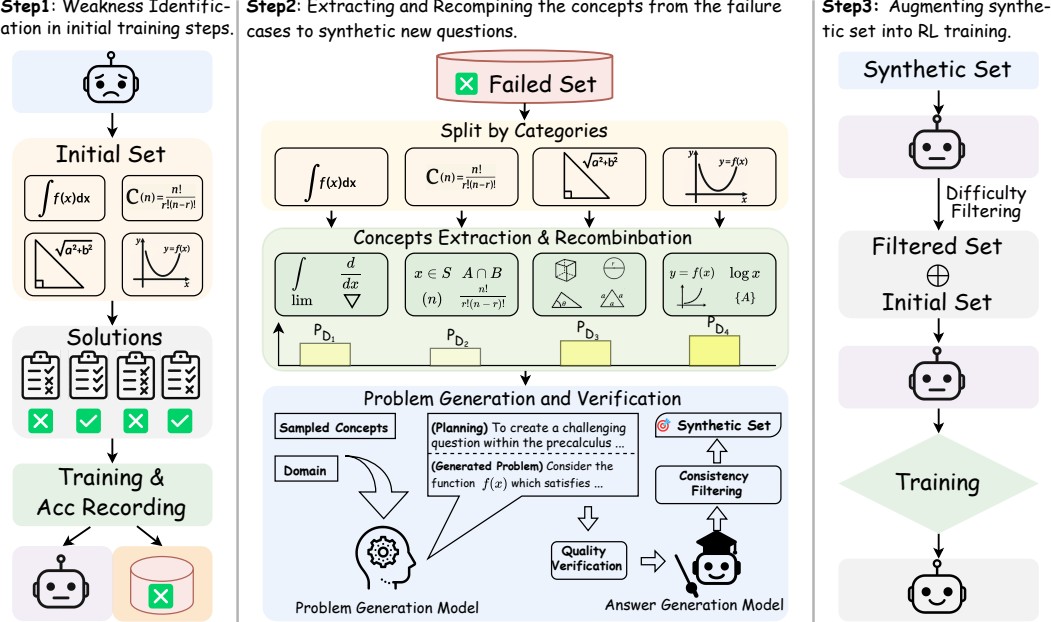

Figure 3: An overview of our proposed weakness-driven problem synthesis framework that targets at mitigating the model's reasoning limitations within the RLVR paradigm.

## 2.2 Overview

Figure 3 presents an overview of our SwS framework, which generates targeted training samples to enhance the model's reasoning capabilities in RLVR. The framework initiates with a *Self-aware Weakness Identification* stage, where the model undergoes preliminary RL training on an initial problem set covering diverse categories. During this stage, the model's weaknesses are identified as problems it consistently fails to solve or learns ineffectively. Based on failure cases that reflect the model's weakest capabilities, in the subsequent *Targeted Problem Synthesis* stage, we group them by category, extract their underlying concepts, and recombine these concepts to synthesize new problems that target the model's learning and mitigation of its weaknesses. In the final *Augmented Training with Synthetic Problems* stage, the model receives continuous training with the augmented high-quality synthetic problems, thereby enhancing its general reasoning abilities through more targeted training.

## 2.3 Self-aware Weakness Identification

Utilizing the policy model itself to identify its weakest capabilities, we begin by training it in a preliminary RL phase using an initial problem set $\mathbf{X}_S$, which consists of mathematical problems from $n$ diverse categories $\{\mathbf{D}\}_{i=0}^{n}$, each paired with a ground-truth answer $a$. As illustrated in Figure 2, we record the average accuracy $a_{i,t}$ of the model's responses to each prompt $x_i$ at each epoch $t \in \{0, 1, \ldots, T_1\}$, where $T_1$ is the number of training epochs in this phase. We track the **Failure Rate** $F$ for each problem in the training set to identify those that the model consistently struggles to learn, which are considered its weaknesses. Specifically, such problems are defined as those the model consistently struggles to solve during RL training, which meet two criteria: (1) The model never reaches a response accuracy of 50% at any training epoch, and (2) The accuracy trend decreases over time, indicated by a negative slope:

$$F(x_i) = \mathbb{I}\left[\max_{t \in [1,T]} a_{i,t} < 0.5 \ \wedge \ \text{slope}\left(\{a_{i,t}\}_{t=1}^{T}\right) < 0\right] \tag{4}$$

This metric captures both problems the model consistently fails to solve and those showing no improvement during sampling-based RL training, making them appropriate targets for training augmentation. After the weakness identification phase via the preliminary training on the initial training set $\mathbf{X}_S$, we employ the collected problems $\mathbf{X}_F = \{x_i \in \mathbf{X}_S \mid F_r(x_i) = 1\}$ as seed problems for subsequent weakness-driven problem synthesis.

## 2.4 Targeted Problem Synthesis

**Concept Extraction and Recombination**. We synthesize new problems by extracting the underlying concepts $\mathbf{C}_F$ from the collected seed questions $\mathbf{X}_F$ and strategically recombining them to generate questions that target similar capabilities. Specifically, the extracted concepts are first categorized into their respective categories $\mathbf{D}_i$ (e.g., mathematical topics such as *Algebra* or *Geometry*) based on the corresponding seed problem $x_i$, and are subsequently sampled and recombined to generate problems within the same category. Inspired by [Huang et al., 2024, Zhao et al., 2025b], we enhance the coherence and semantic fluency of synthetic problems by computing co-occurrence probabilities and embedding similarities among concepts within each category, enabling more appropriate sampling and recombination of relevant concepts. This targeted sampling approach ensures that the synthesized problems remain semantically coherent and avoids combining concepts from unrelated sub-topics or irrelevant knowledge points, which could otherwise result in invalid or confusing questions. Further details on the co-occurrence calculation and sampling algorithm are provided in *Appendix* F.

Intuitively, categories exhibiting more pronounced weaknesses demand additional learning support. To optimize the efficiency of targeted problem synthesis and weakness mitigation in subsequent RL training, we allocate the augmentation budget, i.e., the concept combinations used as inputs for problem synthesis, across categories based on the model's category-specific failure rates $F_{\mathbf{D}}$ from the preliminary training phase. Specifically, we normalize these failure rates $F_{\mathbf{D}}$ across categories to determine the allocation weights for problem synthesis. Given a total augmentation budget $|\mathbf{X}_T|$, the number of concept combinations allocated to domain $\mathbf{D}_i$ is computed as:

$$|\mathbf{X}_{T,\mathbf{D}_i}| = |\mathbf{X}_T| \cdot P_{\mathbf{D}_i} = |\mathbf{X}_T| \cdot \frac{F_{\mathbf{D}_i}}{\sum_j^n F_{\mathbf{D}_j}}, \tag{5}$$

where $F_{\mathbf{D}_i}$ is the failure rate of problems in category $\mathbf{D}_i$ within the initial training set. The sampled and recombined concepts then serve as inputs for subsequent problem generation.

**Problem Generation and Quality Verification**. After extracting and recombining the concepts associated with the model's weakest capabilities, we employ a strong instruction model, which does not perform deep reasoning, to generate new problems based on the category label and the recombined concepts. We instruct the model to first generate rationales that explore how the concept combinations can be integrated to produce a well-formed problem. To ensure the synthetic problems align with the RLVR setting, the model is also instructed to avoid generating multiple-choice, multi-part, or proof-based questions [Albalak et al., 2025]. Detailed prompt used for the concept-based problem generation please refer to the *Appendix* K. For quality verification of the synthetic problems, we prompt general instruction LLMs multiple times to evaluate each problem and its rationale across multiple dimensions, including *concept coverage*, *factual accuracy*, and *solvability*, assigning an overall rating of *bad*, *acceptable*, or *perfect*. Only problems receiving 'perfect' ratings above a predefined threshold and no 'bad' ratings are retained for subsequent utilization.

**Reference Answer Generation**. Since alignment between the model's final answer and the reference answer is the primary training signal in RLVR, a rigorous verification of the reference answers for synthetic problems is essential to ensure training stability and effectiveness. To this end, we employ a strong reasoning model (e.g., QwQ-32B [Team, 2025]) to label reference answers for synthetic problems through a self-consistency paradigm. Specifically, we prompt it to generate multiple responses for each problem and use Math-Verify to assess answer equivalence, which ensures that consistent answers of different forms (e.g., fractions and decimals) are correctly recognized as equal. Only problems with at least 50% consistent answers are retained, as highly inconsistent answers are unreliable as ground truth and may indicate that the problems are excessively complex or unsolvable.

**Difficulty Filtering**. The most prevalently used RLVR algorithms, such as GRPO, compute the advantage of each token in a response by comparing its reward to those of other responses for the same prompt. When all responses yield identical accuracy—either all correct or all incorrect—the advantages uniformly degrade to zero, leading to gradient vanishing for policy updates and resulting in training inefficiency [Shao et al., 2024, Yu et al., 2025b]. Recent study [Wen et al., 2025] further shows that RLVR training can be more efficient with problems of appropriate difficulty. Considering this, we select synthetic problems of appropriate difficulty based on the initially trained model's accuracy on them. Specifically, we sample multiple responses per synthetic problem using the initially trained model and retain only those whose accuracy falls within a target range $[\text{acc}_{\text{low}}, \text{acc}_{\text{high}}]$ (e.g., $[25\%, 75\%]$). This strategy ensures that the model engages with learnable problems, enhancing both the stability and efficiency of RLVR training.

| Model | GSM8K | MATH 500 | Minerva Math | Olympiad Bench | GaoKao 2023 | AMC23 | AIME24 (Avg@ 1 / 32) | AIME25 (Avg@ 1 / 32) | Avg. |
|---|---|---|---|---|---|---|---|---|---|
| Qwen 2.5 3B Base | | | | | | | | | |
| Qwen2.5-3B | 69.9 | 46.0 | 18.8 | 19.9 | 34.8 | 27.5 | 0.0 / 2.2 | 0.0 / 1.5 | 27.1 |
| Qwen2.5-3B-IT | 84.2 | 62.2 | 26.5 | 27.9 | 53.5 | 32.5 | 6.7 / 5.0 | 0.0 / 2.3 | 36.7 |
| BaseRL-3B | 86.3 | 66.0 | 25.4 | 31.3 | 57.9 | 40.0 | 10.0 / 9.9 | 6.7 / 3.5 | 40.4 |
| SwS-3B | 87.0 | 69.6 | 27.9 | 34.8 | 59.7 | 47.5 | 10.0 / 8.4 | 6.7 / 7.1 | 42.9 |
| Δ | +0.7 | +3.6 | +2.5 | +3.5 | +1.8 | +7.5 | +0.0 / -1.5 | +0.0 / +3.6 | **+2.5** |
| LLaMA 3.1 8B Instruct | | | | | | | | | |
| LLaMA-3.1-8B-IT | 85.6 | 48.2 | 24.6 | 18.8 | 39.7 | 22.5 | 6.7 / 3.1 | 3.3 / 2.2 | 31.1 |
| Baseline RL | 88.3 | 58.4 | 31.2 | 23.4 | 49.6 | 30.0 | 16.7 / 9.8 | 6.7 / 5.0 | 38.0 |
| SwS-LLaMA-8B | 90.5 | 60.2 | 33.5 | 25.8 | 49.1 | 40.0 | 16.7 / 11.2 | 6.7 / 6.8 | 40.3 |
| Δ | +2.2 | +1.8 | +2.3 | +2.4 | -0.5 | +10.0 | +0.0 / +1.4 | +0.0 / +1.8 | **+2.3** |
| Qwen 2.5 7B Base | | | | | | | | | |
| Qwen2.5-7B | 88.1 | 63.0 | 27.6 | 30.5 | 55.8 | 35.0 | 6.7 / 5.4 | 0.0 / 1.2 | 38.3 |
| Qwen2.5-7B-IT | 91.7 | 75.6 | 38.2 | 40.6 | 63.9 | 50.0 | 16.7 / 10.5 | 13.3 / 6.7 | 48.8 |
| Open-Reasoner-7B | 93.6 | 80.4 | 39.0 | 45.6 | 72.0 | 72.5 | 10.0 / 16.8 | 13.3 / 17.9 | 53.3 |
| SimpleRL-Base-7B | 90.8 | 77.2 | 35.7 | 41.0 | 66.2 | 62.5 | 13.3 / 14.8 | 6.7 / 6.7 | 49.2 |
| BaseRL-7B | 92.0 | 78.4 | 36.4 | 41.6 | 63.4 | 45.0 | 10.0 / 14.5 | 6.7 / 6.5 | 46.7 |
| SwS-7B | 93.9 | 82.6 | 41.9 | 49.6 | 71.7 | 67.5 | 26.7 / 18.3 | 20.0 / 18.5 | 56.7 |
| Δ | +1.9 | +4.2 | +5.5 | +8.0 | +8.3 | +22.5 | +16.7 / +3.8 | +13.3 / +12.0 | **+10.0** |
| Qwen 2.5 7B Math | | | | | | | | | |
| Qwen2.5-Math-7B | 43.2 | 72.0 | 35.7 | 17.6 | 31.4 | 47.5 | 10.0 / 9.4 | 0.0 / 2.9 | 32.2 |
| Qwen2.5-Math-7B-IT | 93.3 | 80.6 | 36.8 | 36.6 | 64.9 | 45.0 | 6.7 / 7.2 | 13.3 / 6.2 | 47.2 |
| PRIME-RL-7B | 93.2 | 82.0 | 41.2 | 46.1 | 67.0 | 60.0 | 23.3 / 16.1 | 13.3 / 16.2 | 53.3 |
| SimpleRL-Math-7B | 89.8 | 78.0 | 27.9 | 43.4 | 64.2 | 62.5 | 23.3 / 24.5 | 20.0 / 15.6 | 51.1 |
| Oat-Zero-7B | 90.1 | 79.4 | 38.2 | 42.4 | 67.8 | 70.0 | 43.3 / 29.3 | 23.3 / 11.8 | 56.8 |
| BaseRL-Math-7B | 90.2 | 78.8 | 37.9 | 43.6 | 64.4 | 57.5 | 26.7 / 23.0 | 20.0 / 14.0 | 51.9 |
| SwS-Math-7B | 91.9 | 83.8 | 41.5 | 47.7 | 71.4 | 70.0 | 33.3 / 25.9 | 26.7 / 18.2 | 58.3 |
| Δ | +1.7 | +5.0 | +3.6 | +4.1 | +7.0 | +12.5 | +6.7 / +2.9 | +6.7 / +4.2 | **+6.4** |
| Qwen 2.5 32B base | | | | | | | | | |
| Qwen2.5-32B | 90.1 | 66.8 | 34.9 | 29.8 | 55.3 | 50.0 | 10.0 / 4.2 | 6.7 / 2.5 | 42.9 |
| Qwen2.5-32B-IT | 95.6 | 83.2 | 42.3 | 49.5 | 72.5 | 62.5 | 23.3 / 15.0 | 20.0 / 13.1 | 56.1 |
| Open-Reasoner-32B | 95.5 | 82.2 | 46.3 | 54.4 | 75.6 | 57.5 | 23.3 / 23.5 | 33.3 / 31.7 | 58.5 |
| SimpleRL-Base-32B | 95.2 | 81.0 | 46.0 | 47.4 | 69.9 | 82.5 | 33.3 / 26.2 | 20.0 / 15.0 | 59.4 |
| BaseRL-32B | 96.1 | 85.6 | 43.4 | 54.7 | 73.8 | 85.0 | 40.0 / 30.7 | 6.7 / 24.6 | 60.7 |
| SwS-32B | 96.3 | 89.4 | 47.1 | 60.5 | 80.3 | 90.0 | 43.3 / 33.0 | 40.0 / 31.8 | 68.4 |
| Δ | +0.2 | +3.8 | +3.7 | +5.8 | +6.5 | +5.0 | +3.3 / +2.3 | +33.3 / +7.2 | **+7.7** |

Table 1: We report the detailed performance of our SwS implementation across various base models and multiple benchmarks. AIME is evaluated using two metrics: Avg@1 (single-run performance) and Avg@32 (average over 32 runs).

## 2.5 Augmented Training with Synthetic Problems

After the rigorous problem generation, answer generation, and verification, the allocation budget of synthetic problems in each category is further adjusted using the weights in Eq. 5 to ensure their comprehensive and efficient utilization, resulting in $\mathbf{X}'_T$. We incorporate the retained synthetic problems $\mathbf{X}'_T$ into the initial training set $\mathbf{X}_S$, forming the augmented training set $\mathbf{X}_A = [\mathbf{X}_S; \mathbf{X}'_T]$. We then continue training the initially trained model on $\mathbf{X}_A$ in a second stage of augmented RLVR, targeting to mitigate the model's weaknesses through exploration of the synthetic problems.

## 3 Experiments

### 3.1 Experimental Setup

**Models and Datasets**. We employ the Qwen2.5-base series [Yang et al., 2024a,b] with model sizes from 3B to 32B in our experiments. To further demonstrate the generalizability of our method, we also adopt the LLaMA-3.1-8B-Instruct[Grattafiori et al., 2024] model for SwS data augmentation. For concept extraction and problem generation, we employ the LLaMA-3.3-70B-Instruct model [Grattafiori et al., 2024], and for concept embedding, we use the LLaMA-3.1-8B-base model. To verify the quality of the synthetic questions, we use both the LLaMA-3.3-70B-Instruct and additionally Qwen-2.5-72B-Instruct [Yang et al., 2024a] to evaluate them and filter out the low-quality samples. For answer generation, we use Skywork-OR1-Math-7B [He et al., 2025] for training models with sizes up to 7B, and QwQ-32B [Team, 2025] for the 32B model experiments. We employ the SwS pipeline to generate 40k synthetic problems for each base model. All the prompts for each procedure

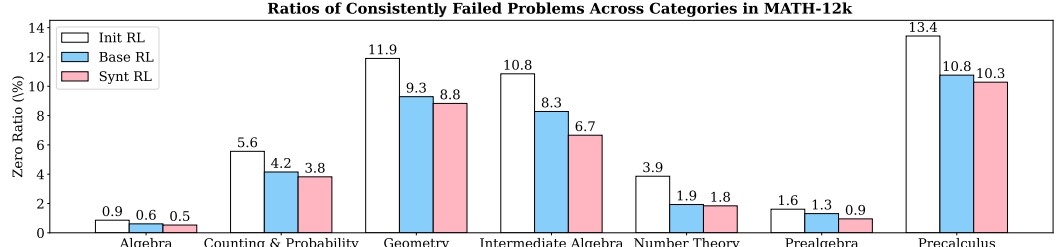

Figure 4: The ratios of consistently failed problems from different categories in the MATH-12k training set under different training configurations. (Base model: Qwen2.5-7B).

|  | Weakness Identification | Concepts Extraction | Problem Generation | Quality Verification | Answer Generation | Difficulty Filtering | Augmented Training |
|---|---|---|---|---|---|---|---|
| GPU Hours (h) | 2,075 | 1.6 | 1,940 | 975 | 4,608 | 768 | 9,543 |
| Data Quantity | 12k | 1,339 | 1,000k | 842k | 302k | 253k | 52k |
| Data / Hour | - | 836.9 | 515.5 | 863.6 | 65.5 | 329.4 | - |

Table 2: Stage-wise GPU Hours for SwS Experiments with Qwen2.5-7B.

in SwS can be found in Appendix K. We adopt GRPO [Shao et al., 2024] as the RL algorithm, and full **implementation details** are in Appendix C.

For the initial training set used in the preliminary RL training for weaknesses identification, we employ the MATH-12k [Hendrycks et al., 2021] for models with sizes up to 7B. As the 14B and 32B models show early saturation on MATH-12k, we instead use a combined dataset of 17.5k samples from the DAPO [Yu et al., 2025b] English set and the LightR1 [Wen et al., 2025] Stage-2 set.

**Evaluation**. We evaluated the models on a wide range of mathematical reasoning benchmarks, including GSM8K [Cobbe et al., 2021], MATH-500 [Lightman et al., 2023], Minerva Math [Lewkowycz et al., 2022], Olympiad-Bench [He et al., 2024], Gaokao-2023 [Zhang et al., 2023], AMC [MAA, a], and AIME [MAA, b]. We report Pass@1 (Avg@1) accuracy across all benchmarks and additionally include the Avg@32 metric for the competition-level AIME benchmark to enhance evaluation robustness. For detailed descriptions of the evaluation benchmarks, see Appendix J.

**Baseline Setting**. Our baselines include the base model, its post-trained Instruct version (e.g., Qwen2.5-7B-Instruct), and the initial trained model further trained on the initial dataset for the same number of steps as our augmented RL training as the baselines. To further highlight the effectiveness of the SwS framework, we compare the model trained on the augmented problem set against recent advanced RL-based models, including SimpleRL [Zeng et al., 2025], Open Reasoner [Hu et al., 2025], PRIME [Cui et al., 2025], and Oat-Zero [Liu et al., 2025b].

## 3.2 Main Results

The overall experimental results are presented in Table 1. Our SwS framework enables consistent performance improvements across benchmarks of varying difficulty and model scales, with the most significant gains observed in models greater than 7B parameters. Specifically, SwS-enhanced versions of the 7B and 32B models show absolute improvements of +10.0% and +7.7%, respectively, underscoring the effectiveness and scalability of the framework. When initialized with MATH-12k, SwS yields strong gains on competition-level benchmarks, achieving +16.7% and +13.3% on AIME24 and AIME25 with Qwen2.5-7B. These results highlight the quality and difficulty of the synthesized samples compared to well-crafted human-written ones, demonstrating the effectiveness of generating synthetic data based on model capabilities to enhance training.

## 3.3 Weakness Mitigation from Augmented Training

The motivation behind SwS is to mitigate model weaknesses by explicitly targeting failure cases during training. To demonstrate its effectiveness, we use Qwen2.5-7B to analyze the ratios of consistently failed problems in the initial training set (MATH-12k) across three models: the initially

| Model | GSM8K | AIME24 (Pass@32) | Prealgebra | Intermediate Algebra | Algebra | Precalculus | Number Theory | Counting & Probability | Geometry |
|---|---|---|---|---|---|---|---|---|---|
| Strong Student | 92.0 | 13.8 | 87.7 | 58.7 | 93.8 | 63.2 | 86.4 | 71.2 | 66.8 |
| Weak Teacher | 93.3 | 7.2 | 88.2 | 64.3 | 95.5 | 71.2 | 93.0 | 81.4 | 63.0 |
| Trained Student | 93.6 | 17.5 | 90.5 | 64.4 | 97.7 | 74.6 | 95.1 | 80.4 | 67.5 |

Table 3: Performance on two representative benchmarks and category-specific results on MATH-500 of the weak teacher model and the strong student model.

| Model | GSM8K | MATH 500 | Minerva Math | Olympiad Bench | GaoKao 2023 | AMC23 | AIME24 (Avg@ 1 / 32) | AIME25 (Avg@ 1 / 32) | Avg. |
|---|---|---|---|---|---|---|---|---|---|
| Qwen2.5-14B-IT | 94.7 | 79.6 | 41.9 | 45.6 | 68.6 | 57.5 | 16.7 / 11.6 | 6.7 / 10.9 | 51.4 |
| + BaseRL | 94.5 | 85.4 | 44.1 | 52.1 | 71.7 | 65.0 | 20.0 / 21.6 | 20.0 / 22.3 | 56.6 |
| + SwS-SE | 95.6 | 85.0 | 46.0 | 53.5 | 74.8 | 67.5 | 20.0 / 19.8 | 20.0 / 17.8 | 57.8 |
| Δ | +1.1 | -0.4 | +1.9 | +1.4 | +3.1 | +2.5 | +0.0 / -1.8 | +0.0 / -4.5 | +1.2 |

Table 4: Experimental results of extending the SwS framework to the *Self-evolving* paradigm on the Qwen2.5-14B-Instruct model.

trained model, the model continued trained on the initial training set, and the model trained on the augmented set with synthetic problems from the SwS pipeline. As shown in Figure 4, continued training on the augmented set enables the model to solve a greater proportion of previously failed problems across most domains compared to training on the initial set alone, with the greatest gains observed in *Intermediate Algebra* (20%), *Geometry* (5%), and *Precalculus* (5%) as its weakest areas. Notably, these improvements are achieved even though each original problem is sampled four times less frequently in the augmented set than in training on the original dataset alone, highlighting the efficiency of SwS-generated synthetic problems in RL training.

### 3.4 GPU Hours Analysis for SwS

For the specific GPU hours at each stage, we use the Qwen2.5 7B experiment as an example and report the GPU hours for each SwS stage in the Table 2. All time measurements are based on NVIDIA A100 40G GPUs. Notably, the total time spent on all problem synthesis stages (8,292.6 GPU hours) is actually less than that required by the final augmented training via RL (9,543 GPU hours). This comparison highlights the rationale and necessity for allocating computational resources to data augmentation prior to RL. Within the problem synthesis pipeline, the most time-consuming component is *Answer Generation*, as it requires a powerful reasoning model to ensure answer correctness. In contrast, other stages mainly involve shorter inference, thus consumes less time.

## 4 Extensions and Analysis

### 4.1 Weak-to-Strong Generalization for SwS

Employing a powerful frontier model like QwQ [Team, 2025] helps ensure answer quality. However, when training the top-performing reasoning model, no stronger model exists to produce reference answers for problems identified as its weaknesses. To explore the potential of applying our SwS pipeline to enhancing state-of-the-art models, we extend it to the *Weak-to-Strong Generalization* [Burns et al., 2023] setting by using a generally weaker teacher that may outperform the stronger model in specific domains to label reference answers for the synthetic problems.

Intuitively, using a weaker teacher may result in mislabeled answers, which could significantly impair subsequent RL training. However, during the *difficulty filtering* stage, this risk is mitigated by using the initially trained policy to assess the difficulty of synthetic problems, as it rarely reproduces the same incorrect answers provided by the weaker teacher. As a byproduct, mislabeled cases are naturally filtered out alongside overly complex samples through accuracy-based screening. The experimental analysis on the validity of difficulty-level filtering in ensuring label correctness is presented in Table 6.

We use the initially trained Qwen2.5-7B-Base as the student and Qwen2.5-Math-7B-Instruct as the teacher. Table 3 presents their performance on popular benchmarks and MATH-12k categories, where

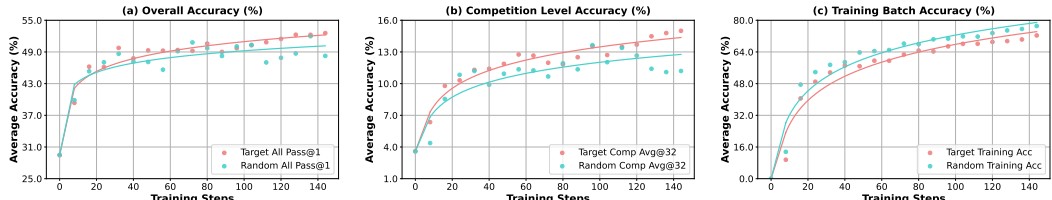

Figure 5: Comparison of accuracy improvements using (a) Pass@1 on full benchmarks evaluated in Table 1 and (b) Avg@32 on the competition-level benchmarks. (c) illustrates the proportion of prompts within a batch that achieved 100% correctness across multiple rollouts during training.

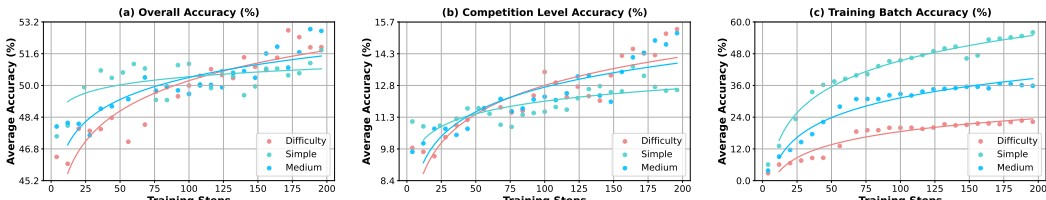

Figure 6: Comparison of incorporating synthetic problems of varying difficulty levels during the augmented RL training. For a detailed description of accuracy trends on evaluation benchmarks and the training set, refer to the caption in Figure 5.

the student model generally outperforms the teacher. However, as shown in Table 3, the student policy further improves after training on weak teacher-labeled problems. This improvement stems from the *difficulty filtering* process, which removes problems with consistent student-teacher disagreement and retains those where the teacher is reliable but the student struggles, enabling targeted training on weaknesses. Detailed analysis can be found in Appendix G.

## 4.2 Self-evolving Targeted Problem Synthesis

In this section, we explore the potential of utilizing the *Self-evolving* paradigm to address model weaknesses by executing the full SwS pipeline using the policy itself. This self-evolving paradigm for identifying and mitigating weaknesses leverages self-consistency to guide itself to generate effective trajectories toward accurate answers [Zuo et al., 2025], while also integrating general instruction-following capabilities from question generation and quality filtering to enhance reasoning.

We use Qwen2.5-14B-Instruct as the base policy due to its balance between computational efficiency and instruction-following performance. The results are shown in Table 4, where the self-evolving SwS pipeline improves the baseline performance by 1.2% across all benchmarks, especially on the middle-level benchmarks like Gaokao and AMC. Although performance declines on AIME, we attribute this to the initial training data from DAPO and LightR1 already being specifically tailored to that benchmark. For further discussion of the *Self-evolve* SwS framework, refer to Appendix H.

## 4.3 Weakness-driven Selection

In this section, we explore an alternative extension that augments the initial training set using identified weaknesses and a larger mathematical reasoning dataset. Specifically, we use the Qwen2.5-7B model, identify its weaknesses on the MATH-12k training set, and retrieve augmented problems from Big-Math [Albalak et al., 2025] that align with its failure cases, incorporating them into the initial training set for augmentation. We employ a category-specific selection strategy similar to the budget allocation in Eq. 5, using KNN [Cover and Hart, 1967] to identify the most relevant problems within each category. The total augmentation budget is also set to 40k. We compare this approach to a baseline where the model is trained on an augmented set incorporated with randomly selected problems from Big-Math. Details of the selection procedure are provided in Appendix I.

As shown in Figure 5, the model trained with weakness-driven augmentation outperforms the random augmentation strategy in terms of accuracy on both the whole evaluated benchmarks (Figure 5.a) and the competition-level subset (Figure 5.b), demonstrating the effectiveness of the weakness-driven

| Original Problem | Synthetic Problems of Diverse Difficulty levels |
|---|---|
| Equilateral $\Delta ABC$ has side length 600. Points $P$ and $Q$ lie outside the plane of $\Delta ABC$ and are on opposite sides of the plane. Furthermore, $PA = PB = PC$, and $QA = QB = QC$, and the planes of $\Delta PAB$ and $\Delta QAB$ form a 120° dihedral angle (the angle between the two planes). There is a point $O$ whose distance from each of $A, B, C, P$, and $Q$ is $d$. Find $d$. | **Simple**: Two cones, $A$ and $B$, are similar, with cone $A$ being tangent to a sphere. The radius of the sphere is $r$, and the height of cone A is $h$. If the ratio of the height of cone $B$ to the height of cone $A$ is $k$, find the ratio of the surface area of cone $B$ to the surface area of cone $A$. **Answer**: $k^2$, **Model Accuracy**: 100% |
| **Extracted Concepts** | **Medium**: In a circle with radius $r$, two tangents are drawn from a point $P$ such that the angle between them is 60°. If the length of each tangent is $r\sqrt{3}$ find the distance from $P$ to the center. **Answer**: $2r$, **Model Accuracy**: 50% |
| Geometric shapes and their properties Properties of equilateral triangles Understanding of points and planes in 3D space Distance and midpoint formulas in 3D space Properties of perpendicular lines and planes | **Hard**: In triangle $ABC$, let $I$ be the incenter and $E$ the excenter opposite $A$. If $AE = 5$, $AI = 3$, and $EI$ is tangent to the incircle at $D$, find the radius. **Answer**: 2, **Model Accuracy**: 6.25% **Unsolvable**: In triangle $ABC$, with $AB = 7$, $AC = 9$, and $\angle A = 60°$, let $D$ be the midpoint of $BC$. Given $BD$ is 3 more than $DC$, find $AD$. **Answer**: 15/2, **Model Accuracy**: 0% |

Figure 7: Illustration of a geometry problem from the MATH-12k failed set, with extracted concepts and conceptually linked synthetic problems across different difficulty levels.

selection strategy. In Figure 5.c, it is worth noting that the model quickly fits the randomly selected problems in training, which then cease to provide meaningful training signals in the GRPO algorithm. In contrast, since the failure cases highlight specific weaknesses of the model's capabilities, the problems selected based on them remain more challenging and more aligned with its deficiencies, providing richer learning signals and promoting continued development of reasoning skills.

## 4.4 Impact of Question Difficulty

We ablate the impact of the difficulty levels of synthetic problems used in the augmented RL training. In this section, we define the difficulty of a synthetic problem based on the accuracy of multiple rollouts generated by the initially trained model, base from Qwen2.5-7B. We incorporate synthetic problems of three predefined difficulty levels—simple, medium, and hard—into the augmented RL training. These levels correspond to accuracy ranges of $[5, 7]$, $[3, 5]$, and $[1, 4]$ out of 8 sampled responses, respectively. For each level, we sample 40k examples and combine them with the initial training set for a second training stage lasting 200 steps.

The experimental results are shown in Figure 6. Similar to the findings in Section 4.3, the model fits more quickly on the simple augmented set and initially achieves the best performance across all evaluation benchmarks, including competition-level tasks, but then saturates with no further improvement. In contrast, the medium and hard augmented sets lead to slower convergence on the training set but result in more sustained performance gains on the evaluation set, with the hardest problems providing the longest-lasting training benefits.

## 4.5 Case Study

Figure 7 presents an illustration of a geometry failure case from the MATH-12k training set, accompanied by extracted concepts and our weakness-driven synthetic questions of varying difficulty levels, all closely aligned with the original question. The question focuses on three-dimensional distance and triangle understanding, with key concepts such as "Properties of equilateral triangles" and "Distance and midpoint formulas in 3D space" representing essential knowledge required to solve the problem. Notably, the corresponding synthetic questions exhibit similar semantics—such as "finding distance" in Medium and "understanding triangles" in Hard. Practicing on such targeted problems helps mitigate weaknesses and enhances reasoning capabilities within the relevant domain.

## 5 Conclusion

In this work, we introduce a Self-aware Weakness-driven Problem Synthesis (SwS) framework (SwS) in reinforcement learning for LLM reasoning, which synthesizes problems based on weaknesses identified from the model's failure cases during a preliminary training phase and includes them into subsequent augmented training. We conduct a detailed analysis of incorporating such synthetic problems into training and find that focusing on the model's failures can enhance its reasoning generalization and mitigate its weaknesses, resulting in overall performance improvements. Furthermore, we extend the framework to the paradigms of *Weak-to-Strong Generalization*, *Self-evolving*, and *Weakness-driven Selection*, demonstrating its comprehensiveness and robustness.

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

# Appendix Contents for SwS

# A   Discussions, Limitations and Future Work

This paper presents a comprehensive Self-aware Weakness-driven Problem Synthesis (SwS) framework to address the model's reasoning deficiencies through reinforcement learning (RL) training. Although the SwS framework is effective across a wide range of model sizes, there are still several limitations to it: (1) Employing both a strong instruction model and an answer-labeling reasoning model may lead to computation and time costs. (2) Our framework mainly focuses on the RL setting, as our primary goal is to mitigate the model's weaknesses by fully activating its inherent reasoning abilities without distilling external knowledge. Exploring how to leverage a similar pipeline for enhancing model capabilities through fine-tuning or distillation remains an open direction for future research. (3) The synthetic problems generated by open-source instruction models in the SwS framework may still lack sufficient complexity to elicit the deeper reasoning capabilities of the model, especially on more challenging problems. This limitation is pronounced in the *Self-evolving* setting in Section 4.2, which relies solely on a 14B model for problem generation, with performance improvements limited to only moderate or simple benchmarks. This raises questions about the actual utility of problems generated from the LLaMA-3.3-70B-Instruct in the main experiments on top-challenging benchmarks like AIME. One potential strategy is to use Evolve-Instruct Xu et al. [2023], Luo et al. [2023] to further refine the generated problems to the desired level of difficulty. However, how to effectively raise the upper bound of difficulty in synthetic problems generated by instruction models remains an open problem and warrants further exploration.

In the future, we aim to identify model weaknesses from multiple perspectives beyond simple answer accuracy, with the goal of synthesizing more targeted problems to improve sample efficiency. Additionally, we plan to extend the SwS framework to more general tasks beyond reasoning, incorporating an off-the-shelf reward model to provide feedback instead of verifiable answers. Lastly, we also seek to implement the SwS pipeline in more advanced reasoning models equipped with Long-CoT capabilities, further pushing the boundaries of open-source large reasoning models.

# B   Related Work

Recent advancements have significantly enhanced the integration of reinforcement learning (RL) with large language models (LLMs)[Ziegler et al., 2019, Ouyang et al., 2022], particularly in the domains of complex reasoning and code generation[Guo et al., 2025]. Algorithms such as Proximal Policy Optimization (PPO)[Schulman et al., 2017] and Generalized Reinforcement Preference Optimization (GRPO)[Shao et al., 2024] have demonstrated strong generalization and effectiveness in these applications. In contrast to supervised fine-tuning (SFT) via knowledge distillation Kang et al. [2023], Zhang et al. [2024b], Yu et al. [2025a], RL optimizes a model's reason capabilities on its own generated outputs through reward-driven feedback, thereby prompting stronger generalization. In contrast, SFT models often depend on rote memorization of reasoning patterns and solutions [Chu et al., 2025], and may produce correct answers with flawed rationales [Wang et al., 2025]. In LLM reasoning, RL strengthens policy exploration and improves reasoning performance by using the verified correctness of the final answer in the responses as reward signals for training [Luong et al., 2024], which is commonly referred to as reinforcement learning with verifiable rewards (RLVR) [Yue et al., 2025].

**Robust RLVR for LLM Reasoning**. Scaling up reinforcement learning for LLMs poses significant challenges in terms of training stability and efficiency. Designing stable and efficient supervision algorithms and frameworks for LLMs has attracted widespread attention from the research community.

To address the challenge of reward sparsity in reinforcement learning, recent studies have explored not only answer-based rewards but also process-level reward modeling [Cobbe et al., 2021, Lightman et al., 2023, Wang et al., 2023, Zhang et al., 2025], enabling the provision of more fine-grained reward signals throughout the entire solution process [Wu et al., 2023]. Wang et al. [2023] successfully incorporated a process reward model (PRM), trained on process-level labels generated via Monte Carlo sampling at each step, into RL training and demonstrated its effectiveness. Beyond RL training, PRM can also be used to guide inference [Cobbe et al., 2021] and provide value estimates incorporated with search algorithms [Zhang et al., 2024a, Guan et al., 2025]. However, Guo et al. [2025] found that the scalability of process-level RL is limited by the ambiguous definition of "step" and the high cost of process-level labeling. How to effectively scale process-level RL remains an open question.

Recent efforts in scaling up RLVR optimization have focused on enhancing exploration [Yu et al., 2025b, Yuan et al., 2025, Liu et al., 2025b, Yeo et al., 2025] and adapting RL to the Long-CoT conditions [Jaech et al., 2024, Guo et al., 2025, Li et al., 2025c]. Yu et al. [2025b] found that the KL constraint may limit exploration under RLVR, while Liu et al. [2025b] proposed removing variance normalization in GRPO to prevent length bias. Building on PPO, Yuan et al. [2025] found that pre-training the value function prior to RL training and employing a length-adaptive GAE can improve training stability and efficiency in RLVR, preventing it from degrading to a constant baseline in value estimation.

**Data Construction in RLVR**. Although RL training on simpler mathematical questions can partially elicit a model's reasoning ability [Zeng et al., 2025], the composition of RL training data is critical for enhancing the model's reasoning capabilities [Luo et al., 2025, Yu et al., 2025b, Li et al., 2025a, Hu et al., 2025, He et al., 2025, Shen et al., 2025]. Carefully designing a problem set with difficulty levels matched to the model's abilities and sufficient diversity can significantly improve performance. In addition, the use of curriculum learning has been shown to improve the efficiency of reinforcement learning [Shi et al., 2025]. In this work, we propose generating synthetic problems based on the model's weaknesses for RL training, where the synthetic problems are tailored to align with the model's capabilities and target its areas of weakness, fostering its exploration and improving performance.

**Data Synthesis for LLM Reasoning** Existing data synthesis strategies for enhancing LLM reasoning primarily concentrate on generating problem-response pairs [Huang et al., 2024, Tang et al., 2024, Yu et al., 2023, Zhao et al., 2025b, Liang et al., 2024, Luo et al., 2023, Liu et al., 2025a, Wang et al., 2024, Li et al., 2024b, Tan et al., 2024, Pei et al., 2025] or augmenting responses to existing questions [Toshniwal et al., 2024, Tong et al., 2024, He et al., 2025, Face, 2025, Wen et al., 2025, Yu et al., 2025c, Li et al., 2025b], typically by leveraging advanced LLMs to produce these synthetic examples. A prominent line of work focuses on extracting and recombining key concepts from seed problems. KP-Math [Huang et al., 2024] and MathScale [Tang et al., 2024] decompose seed problems into underlying concepts and recombine them to create new problems, leveraging advanced models to generate corresponding solutions. PromptCoT [Zhao et al., 2025b] also leverages underlying concepts, but focuses on generating competition-level problems. DART-Math [Tong et al., 2024] introduces a difficulty-aware framework that prioritizes the diversity and richness of synthetic responses to challenging problems.

Recently, several studies have emerged aiming to construct distilled datasets to better elicit the reasoning capabilities of LLM. [Guo et al., 2025]. Several works [Face, 2025, Ye et al., 2025, Muennighoff et al., 2025, Lu et al., 2025, Zhao et al., 2025a] employ advanced Long-CoT models to generate responses for distilling knowledge into smaller models. However, a significant disparity in capabilities between the teacher and student models can lead to hallucinations in the student's outputs [Nguyen et al., 2025] and hinder generalization to out-of-distribution scenarios [Chu et al., 2025]. In contrast, our framework under the RL setting enables the model to identify and mitigate its own weaknesses by generating targeted synthetic problems from failure cases, thereby encouraging more effective self-improvement based on its specific weaknesses.

## C Implementation Details

### C.1 Training

We conduct our experiments using the verl [Sheng et al., 2024] framework and adopt GRPO [Shao et al., 2024] as the optimization algorithm. For all RL training experiments, we sample 8 rollouts per problem and use a batch size of 1024, with the policy update batch size set to 256. We employ a constant learning rate of $5 \times 10^{-7}$ with a 20-step warm-up, and set the maximum prompt and response lengths to 1,024 and 8,192 tokens, respectively. We do not apply a KL penalty, as recent studies have shown it may hinder exploration and potentially cause training collapse [Yuan et al., 2025, Liu et al., 2025b, Yu et al., 2025b]. In the initial training stage, we train the model for 200 steps. During augmented RL training, we continually train the initially trained model for 600 steps on the augmented dataset incorporated with synthetic problems, using only prompts with an accuracy between $\text{acc}_{\text{lower}} = 10\%$ and $\text{acc}_{\text{upper}} = 90\%$ as determined by the online policy model for updates. The probability ratio clipping ranges in Eq. 3 is set to $\varepsilon = 0.20$ and $\varepsilon^h = 0.28$.

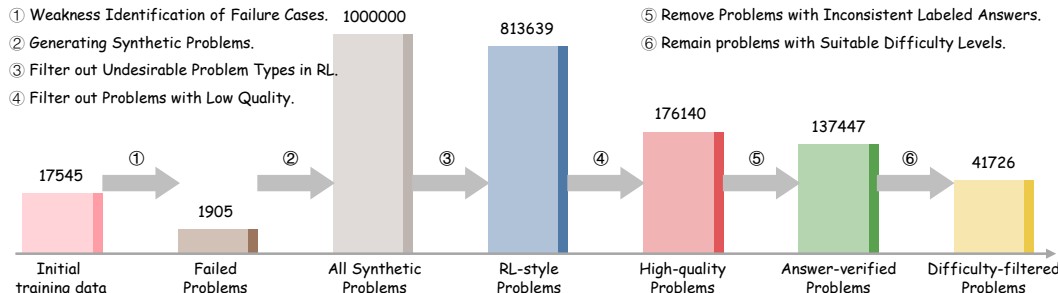

Figure 8: Demonstration of the SwS data workflow by tracing the process from initial training data to the final selection of synthetic problems in the 32B model experiments. For better visualization, the bar heights are scaled using the cube root of the raw data.

Since the training data for the 32B and 14B models (a combination of DAPO [Yu et al., 2025b] and LightR1 [Wen et al., 2025] subsets) lack human-annotated category information, we leverage the LLaMA-3.3-70B-Instruct model to label their categories. This ensures consistency with our SwS pipeline, which combines concepts within the same category. The prompt is presented in Prompt 1.

## C.2  Evaluation

For evaluation, we utilize the vLLM framework [Kwon et al., 2023] and allow for responses up to 8,192 tokens. For all the benchmarks, Pass@1 is computed using greedy decoding for baseline models and sampling (temperature 1.0, top-p 0.95) for RL-trained models. For Avg@32 on competition-level benchmarks, we sample 32 responses per model with the same sampling configuration as used in RL training. We adopt a hybrid rule-based verifier by integrating *Math-Verify* and the PRIME-RL verifier [Cui et al., 2025], as their complementary strengths lead to higher recall. For all the inference, we use the default chat template and enable CoT prompting by appending the instruction: "Let's think step by step and output the final answer within "\boxed{}" after each question.

## D  Motivation for Using RL in Weakness Identification

In our SwS framework, we propose utilizing an initial RL training phase for weakness identification. However, one might argue that there are simpler alternatives for weakness identification, such as directly sampling training problems from the base model or applying supervised fine-tuning before prompting the model to answer questions. In this section, we provide an in-depth discussion on the validity of using problems with low training efficiency during the initial RL phase as model's weaknesses.

We first compare the performance of the Base model, SFT model, and Initial RL model by sampling on the training set, where the SFT model is obtained by fine-tuning the Base model for 1 epoch on human-written solutions. For each question, we prompt the model to generate 8 responses and report the proportion of problems for which none of the responses are correct in Figure 9. For the Base model, failures may be attributed to its insufficient alignment with reasoning-specific tasks. Results from the initial RL model show that the Base model can quickly master such questions through RL, indicating that they do not represent challenging weaknesses. Furthermore, the heavy reliance on the prompt template of the Base model Liu et al. [2025b] reduces its robustness of weakness identification. For the SFT model, there are three main drawbacks regarding weakness identification: (1) The dilemma of training epochs—too many epochs leads to memorizing labeled solutions, while too few epochs fails to align the model with the target problem distribution; (2) SFT is prone to hallucination [Chu et al., 2025, Wang et al., 2025]; and (3) Ensuring the quality of labeled solutions is difficult, as human-written solutions may not always be the best for models Guo et al. [2025]. For these reasons, the SFT model performs poorly on the initial training set, even yielding worse results than the Base model, let alone in utilizing its failed problems to identify model weaknesses.

In contrast to the Base and SFT models, the Initial RL model exhibits the most robust performance on the initial training set, indicating that the failed problems expose the model's most critical weaknesses. Additionally, the training efficiency on all problems during initial RL can also be recorded for further

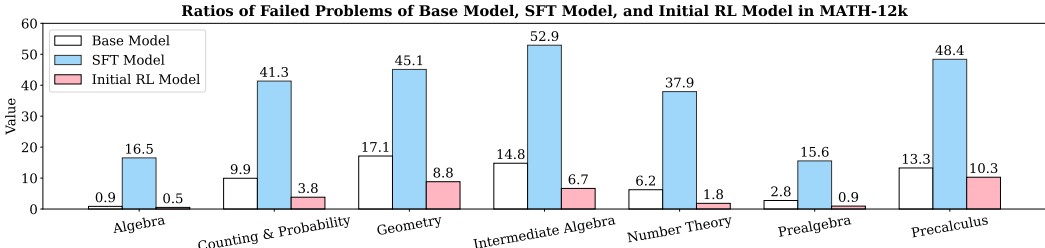

Figure 9: An visualization of utilizing the base model (Qwen2.5-7B), SFT model and the initial RL model on weakness identification in the original training set (MATH-12k).

analysis of model weaknesses. Meanwhile, the initially trained model can also serve as the starting point for augmented RL training. Therefore, in our SwS framework, we ultimately choose to employ an initial RL phase for robust weakness identification.

## E    Data Analysis of the SwS Framework

### E.1    Detailed Data Workflow

Taking the 32B model experiments as an example, Figure 8 shows the comprehensive data workflow of the SwS framework, from identifying model weaknesses in the initial training data to the processing of synthetic problems. The initial training set, consisting of the DAPO and Light-R1 subsets for the Qwen2.5-32B model, contains 17,545 problem-answer pairs. During the weakness identification stage, 1,905 problems are identified as failure cases according to Eq. 4. These failure cases are subsequently used for concept extraction and targeted problem synthesis.

For problem synthesis, we set an initial budget of 1 million synthetic problems in all experiments, with allocations for each category determined as in Eq. 5. These problems then undergo several filtering stages: (1) removing multiple-choice, multi-part, or proof-required problems; (2) discarding problems evaluated as low quality; (3) filtering out problems where the answer generation model yields inconsistent answers, specifically when the most frequent answer among all generations appears less than 50%; and (4) removing problems whose difficulty levels are unsuitable for the current model in RL training. Among these, the quality-based filtering is the strictest, with a filtering rate of 78.35%, indicating that the SwS pipeline maintains rigorous quality control over the generated problems. This ensures both the stability and effectiveness of utilizing synthetic problems in subsequent training.

We present a case study of the quality-based filtering results in Table 5. As illustrated, the positive case that passed the model-based quality evaluation features a concise and precise problem description. In contrast, most synthetic problems identified as low-quality exhibit redundant and overly elaborate descriptions, sometimes including lengthy hints for solving the problem, as seen in the first negative case. Additionally, some low-quality problems incorporate excessive non-mathematical knowledge, such as Physics, as illustrated in the second negative case. The informal *LaTeX* formatting also contributes to their lower quality. Furthermore, problems with multiple question components, such as the third negative case, are also considered as low quality for RL training.

### E.2    Difficulty Distribution of Synthetic Problems

In this section, we study the difficulty distribution of the synthetic problems generated for base models ranging from 3B to 32B, as shown in Figure 10. The red outlines in the pie plots highlight the subset of synthetic problems selected for subsequent augmented RL training, with accuracy falling within the [25%, 75%] range. These samples account for nearly 35% of all generated problems across the four models. The two largest wedges in the pie chart represent problems that the models answered either completely correctly or completely incorrectly. These cases do not provide effective training signals in GRPO [Shao et al., 2024, Yu et al., 2025b], and are thus excluded from the later augmented RL training stage. To further enhance stability and efficiency, we also exclude problems where the model produces only one correct or one incorrect response.

**Positive Case # 1**: Let $z_1$, $z_2$, and $z_3$ be complex numbers such that $|z_1| = |z_2| = |z_3| = 1$ and $z_1 + z_2 + z_3 = 0$. Using the symmetric polynomial $s_2 = z_1 z_2 + z_1 z_3 + z_2 z_3$, find the value of $|s_2|^2$.

**Negative Case # 1**: In a village, there are 10 houses, each of which can be painted one of three colors: red, blue, or green. Two houses cannot have the same color if they are directly adjacent to each other. Using combinatorial analysis and considering the constraints, find the total number of distinct ways to paint the houses, taking into account the possibility of having a sequence where the same color repeats after two different colors (e.g., red, blue, red), and assuming that the color of one of the end houses is already determined to be red, and the colors of the houses are considered different based on their positions (i.e., the configuration red, blue, green is considered different from green, blue, red).

**Negative Case # 2**: A metal's surface requires a minimum energy of 2.5 eV to remove an electron via the photoelectric effect. If light with a wavelength of 480 nm is shone on the metal, and 1 mole of electrons is ejected, what is the total energy, in kilojoules, transferred to the electrons, given that the energy of a photon is related to its wavelength by the formula E $= hc/\lambda$, where $h = 6.626x10^{-34}$ J s and $c = 3.00x10^8 m/s$, and Avogadro's number is $6.02x10^{23}$ particles per mole?

**Negative Case # 3**: In triangle $ABC$, with $\angle A = 60°$, $\angle B = 90°$, $AB = 4$, and $BC = 7$, use the Law of Sines to find $\angle C$ and calculate the triangle's area.

Table 5: Case study of quality filtering results in SwS, featuring one high-quality positive case and three low-quality negative cases. The low-quality segments are marked in pink.

Since all synthetic problems are generated using the same instruction model (LLaMA-3.3-70B-Instruct) with similar competition-level difficulty levels (as illustrated in Prompt 3), and are based on concepts derived from their respective weaknesses, the resulting difficulty distribution of the synthetic problems exhibits only minor differences across all models. Consistent with intuition, the initially trained 3B model achieved the lowest performance on the synthetic questions, with the highest ratio of all-incorrect and the lowest ratio of all-correct responses, while the 32B model showed the opposite trend, achieving the best performance.

## F   Co-occurrence Based Concept Sampling

Following Huang et al. [2024], Zhao et al. [2025b], we enhance the coherence and semantic fluency of synthetic problems by sampling concepts within the same category based on their co-occurrence probabilities and embedding similarities. Specifically, for each candidate concept $c \in \mathbf{C}$ from category $\mathbf{D}$, we define its score based on both co-occurrence statistics and embedding similarity as:

$$\text{Score}(c) = \begin{cases} \text{Co}(c) + \text{Sim}(c), & \text{if } c \notin \{c_1, c_2, \ldots, c_k\} \\ -\infty, & \text{otherwise.} \end{cases}$$

The co-occurrence term $\text{Co}(c)$ is computed by summing the co-occurrence counts from a sparse matrix built over the entire corpus, generated by iterating through all available concept lists in the pool. For each list, we increment $\text{CooccurMatrix}[c, c']$ by one for every unordered pair where $c \neq c'$, yielding a sparse, symmetric matrix in which each entry $\text{CooccurMatrix}[c, c']$ records the total number of times concepts $c$ and $c'$ co-occur across all sampled lists:

$$\text{Co}(c) = \sum_{i=1}^{k} \text{CooccurMatrix}[c, c_i], \tag{6}$$

while the semantic similarity is given by the cosine similarity between the candidate's embedding and the mean embedding of the currently selected concepts:

$$\text{Sim}(c) = \cos\left(\vec{e}_c, \frac{1}{k}\sum_{i=1}^{k} \vec{e}_{c_i}\right), \tag{7}$$

To efficiently support large-scale and high-dimensional concept spaces, we construct a sparse co-occurrence matrix over all unique concepts, where each entry represents the frequency with which a

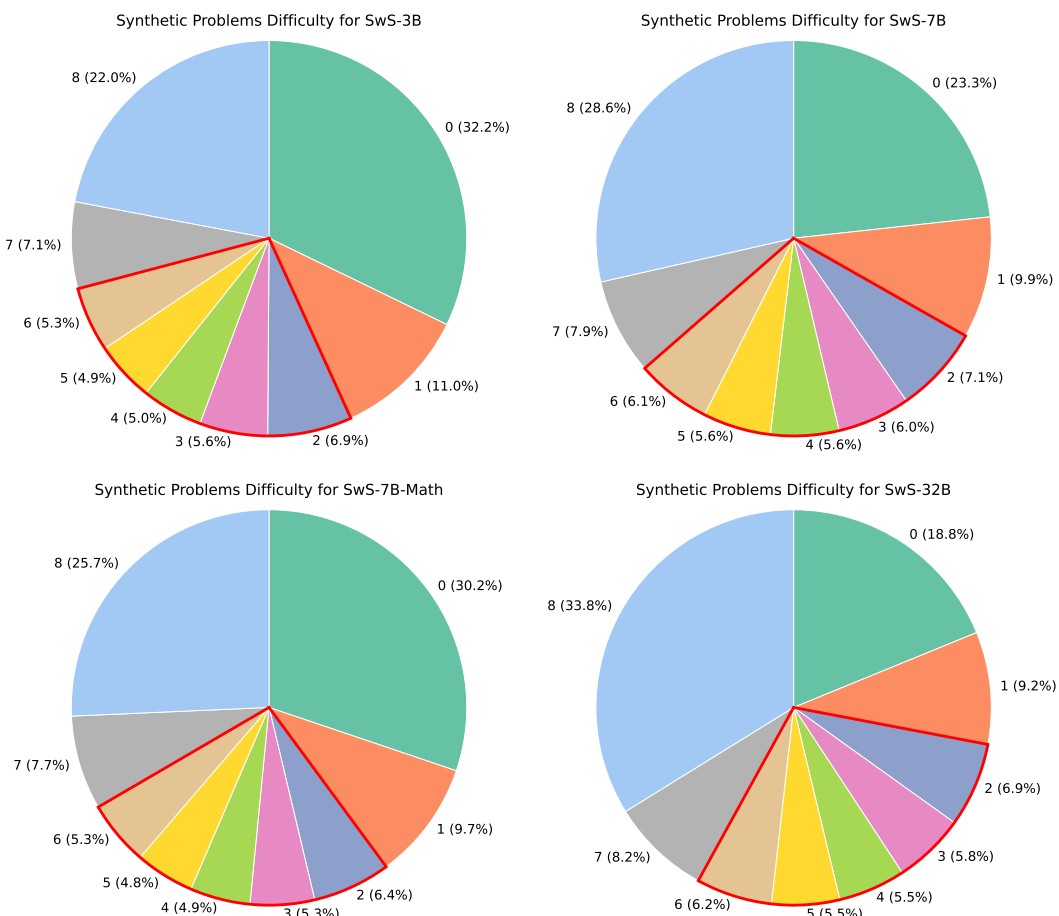

Figure 10: Difficulty distributions of synthetic problems for models from 3B to 32B in our work.

pair of concepts co-occurs within sampled concept lists. Simultaneously, concept embeddings are normalized and indexed via FAISS to facilitate fast similarity computation. During sampling, an initial seed concept is drawn in proportion to its empirical frequency. For each subsequent concept, scores are computed by efficiently summing its co-occurrence with the current set and its embedding similarity to the group mean, while previously selected concepts are masked out. The probability of sampling each candidate is determined via softmax over these scores with temperature $\tau$:

$$P(c) = \frac{\exp\left(\text{Score}(c)/\tau\right)}{\sum_{c' \notin \{c_1, \ldots, c_k\}} \exp\left(\text{Score}(c')/\tau\right)}. \tag{8}$$

This process iteratively constructs coherent, semantically related concept sets to serve as the inputs for synthetic problem generation, ensuring both diversity and fluency.

# G Details for Weak-to-Strong Generalization in SwS

To understand the capabilities of the weak teacher and the strong student model, we evaluated both of them on the MATH-500 test set by prompting them on each question for eight times. Although the teacher model generally exhibits weaker performance, we found that in 16.4% of problems, the weaker teacher outperforms the otherwise stronger student model. This highlights the potential for leveraging a weak teacher to distill its strengths into the student model. A case where the weaker teacher model outperforms the stronger student model is shown in Figure 11.

From the analysis of the SwS framework, as well as its *Weak-to-Strong Generalization* extension, we assert that the upper bound for answer labeling is a revised form of self-consistency score of the teacher model, where (1) the consistent answer must achieve an accuracy greater than 50% across all

**Question:** Compute $99^2 + 99 + 1$ in your head. Please reason step by step, and put your final answer within
`\boxed{}`. **Golden Answer:** $\boxed{9901}$

$>>>$ **Teacher Response:**

To compute $99^2 + 99 + 1$, we can compute each part directly:

$$99^2 = 9801$$
$$99^2 + 99 = 9801 + 99 = 9900$$
$$99^2 + 99 + 1 = 9900 + 1 = 9901$$

Final answer: $\boxed{9901}$

$>>>$ **Student Response:**

Rewrite the expression using a binomial identity:

$$99^2 + 99 + 1 = (99 + 1)^2 = 100^2 = 10000$$

Final answer: $\boxed{10000}$

Figure 11: Case study in which a weak teacher provides the correct solution, while a strong student incorrectly applies a binomial identity and derives an incorrect answer.

| Setting | Size | Prealgebra | Intermediate Algebra | Algebra | Precalculus | Number Theory | Counting & Probability | Geometry | All |
|---|---|---|---|---|---|---|---|---|---|
| Pass@1 | 500 | 88.2 | 64.3 | 95.5 | 71.2 | 93.0 | 81.4 | 63.0 | 80.6 |
| + SC | 500 | 96.9 | 96.0 | 84.4 | 84.1 | 96.2 | 87.5 | 67.8 | 85.4 |
| + SC>50% | 444 | 96.9 | 97.3 | 93.2 | 94.7 | 98.0 | 94.4 | 89.6 | 94.4 |
| + SC>50% & Stu-Con | 407 | 96.8 | 97.2 | 97.7 | 100.0 | 100.0 | 96.8 | 94.9 | 97.5 |

Table 6: The performance of the weak teacher model used for answer generation on the MATH-500 test set under different strategies and their corresponding revisions. "Stu-Con" refers to filtering out problems where the student model's accuracy falls below the defined threshold of 25%.

responses, and (2) the student model must provide the same answer as the teacher model's consistent answer in at least 25% of responses. These revision procedures help ensure the correctness of the synthetic problem answers labeled by the teacher model.

In Table 6, we demonstrate the robustness of utilizing a weaker teacher for answer labeling, assuming that the MATH-500 test set serves as our synthetic problems. As in the second line, even under the self-consistency setting, the teacher model only achieves an improvement of 4.8 points. However, when we exclude problems for which self-consistency does not provide sufficient confidence—specifically, those where the most consistent answer accounts for less than 50% of all responses—the self-consistency setting yields an additional 9.0-point improvement on the remaining questions. Furthermore, in our SwS pipeline, we retain only problems where the student model achieves over 25% accuracy to ensure an appropriate level of difficulty. After filtering out problems where the student falls below this threshold, some mislabeled problems are also automatically removed, resulting in the weak teacher achieving a performance of 97.5% on the final remaining questions. The increase in labeling accuracy from 80.6% to 97.5% shows the potential of utilizing the weaker teacher model for answer labeling as well as the robustness of the SwS framework itself.

# H  Details for Self-Evolving in SwS

As mentioned in Section 4.2, the *Self-evolving* SwS extension enables the policy to achieve better performance on simple to medium-level mathematical reasoning benchmarks but remains suboptimal on AIME-level competition benchmarks. In this section, we further analyze the reasons behind this phenomenon. Figure 12 visualizes the model's self-quality assessment and difficulty evaluation within the SwS framework. Notably, the model assigns a much higher proportion of "perfect" and "acceptable" labels, and fewer "bad" labels, to its self-generated problems compared to the standard framework shown in Figure 8. This observation is consistent with findings from LLM-as-a-Judge [Li et al., 2024b], which indicate that models tend to be more favorable toward and assign higher scores to their own generations. Such behavior may result in overlooking low-quality problems or misclassifying problems that are too complex for the model's reasoning abilities as unsolvable or of poor

**Algorithm 1** Weakness-Driven Selection Pipeline

---

**Require:** Failed Problems $\mathbf{X}_S$; Total Budget $|T|$; Target Set $\mathbf{T}_X$; Domains $\{\mathbf{D}_i\}_{i=0}^n$
**Ensure:** Selected problems $\mathbf{T}_S$
1: **Embed** all failed problems in $\mathbf{X}_S$ and all questions in $\mathbf{T}_X$
2: **for** each domain $\mathbf{D}_i$ in $\{\mathbf{D}_i\}_{i=0}^n$ **do**
3:     **Compute selection budget** $|T_i|$ for $\mathbf{D}_i$ according to Eq. 2
4:     **Extract** failed problems $\mathbf{X}_{S,i}$ belonging to $\mathbf{D}_i$
5:     **for** each $q \in \mathbf{T}_X$ **do**         ▷ Domain-level KNN
6:         Compute $d_i(q) = \min_{f \in \mathbf{X}_{S,i}} \text{distance}(\vec{e}_q, \vec{e}_f)$
7:     **end for**
8:     **Select top** $|T_i|$ questions from $\mathbf{T}_X$ with the smallest $d_i(q)$ as $\mathcal{S}_i$

9: **end for**
10: **return** Selected problems $\mathbf{T}_S = \bigcup_{i=0}^n \mathcal{S}_i$     ▷ Final Selected Set

---

quality. Beyond the risk of filtering out over-complex problems, the model may also have difficulty in accurately labeling answers through self-consistency for over-challenging problems, thereby limiting the potential of incorporating complex problems through the *Self-evolving* SwS framework.

Additionally, in Figure 12, it is noteworthy that the initial RL-trained model achieves nearly 50% all-correct responses on its generated problems, whereas only 31% of problems with appropriate difficulty remain for augmentation after SwS difficulty filtering. This suggests that the self-generated problems may be significantly simpler than those produced using a stronger instruction model [Grattafiori et al., 2024], thus it could lead to data inefficiency and limit the model's performance on more complex problems during RL training.

# I  Details for Weakness-driven Selection

As described in Section 4.3, we utilize the failed problems identified by Qwen2.5-7B [Yang et al., 2024a] on the MATH-12k [Hendrycks et al., 2021] training set, which comprises 915 problems, to select additional data from Big-Math [Albalak et al., 2025] to mitigate the model's weaknesses through the augmented RL training. The complete *Weakness-driven Selection* extension of SwS is presented in Algorithm1. For embedding the problems, we utilize LLaMA-3.1-8B-base [Grattafiori et al., 2024] to encode both the collected failure cases and the problems from the target dataset. The failure cases are then grouped by categories, following the concept sampling strategy in standard SwS. We employ a binary *K-Nearest Neighbors* [Cover and Hart, 1967] algorithm to select weakness-driven problems from the target set, where the augmented problems are chosen by their embedding distances

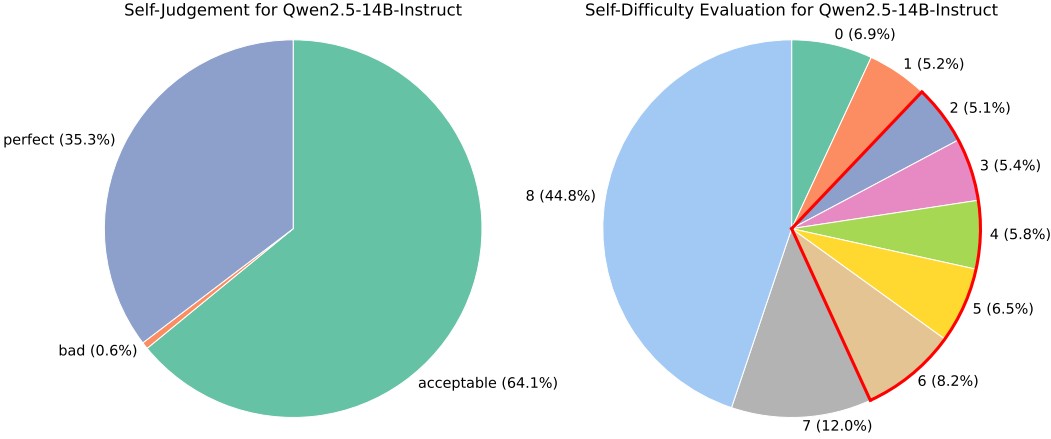

Figure 12: Illustration of the quality assessment and difficulty evaluation for Qwen2.5-14B-Instruct under the *Self-evolving* SwS framework.

to the failure cases within each category. The selection budget for each category is also determined according to Eq.5. We then aggregate the retrieved problems from all categories, forming a selected set of 40k problems, which are then incorporated with the initial set for the subsequent RL training.

## J    Evaluation Benchmark Demonstrations

| Dataset | Size | Category | Example Problem | Answer |
|---------|------|----------|-----------------|--------|
| GSM8k | 1319 | Prealgebra | The ice cream parlor was offering a deal, buy 2 scoops of ice cream, get 1 scoop free. Each scoop cost $1.50. If Erin had $6.00, how many scoops of ice cream should she buy? | 6 |
| MATH-500 | 500 | Geometry | For a constant $c$, in cylindrical coordinates $(r, \theta, z)$, find the shape described by the equation $$z = c.$$ (A) Line (B) Circle (C) Plane (D) Sphere (E) Cylinder (F) Cone. Enter the letter of the correct option. | (C) Plane |
| Minerva Math | 272 | Precalculus | If the Bohr energy levels scale as $Z^2$, where $Z$ is the atomic number of the atom (i.e., the charge on the nucleus), estimate the wavelength of a photon that results from a transition from $n = 3$ to $n = 2$ in Fe, which has $Z = 26$. Assume that the Fe atom is completely stripped of all its electrons except for one. Give your answer in Angstroms, to two significant figures. | 9.6 |
| Olympiad-Bench | 675 | Geometry | Given a positive integer $n$, determine the largest real number $\mu$ satisfying the following condition: for every $4n$-point configuration $C$ in an open unit square $U$, there exists an open rectangle in $U$, whose sides are parallel to those of $U$, which contains exactly one point of $C$, and has an area greater than or equal to $\mu$. | $\frac{1}{2n+2}$ |
| Gaokao2023 | 385 | Geometry | There are three points $A, B, C$ in space such that $AB = BC = CA = 1$. If 2 distinct points are chosen in space such that they, together with $A, B, C$, form the five vertices of a regular square pyramid, how many different ways are there to choose these 2 points? | 9 |
| AMC23 | 40 | Algebra | How many complex numbers satisfy the equation $z^5 = \overline{z}$, where $\overline{z}$ is the conjugate of the complex number $z$? | 7 |
| AIME24 | 30 | Number Theory | Let $N$ be the greatest four-digit positive integer with the property that whenever one of its digits is changed to 1, the resulting number is divisible by 7. Let $Q$ and $R$ be the quotient and remainder, respectively, when $N$ is divided by 1000. Find $Q + R$. | 699 |
| AIME25 | 30 | Geometry | On $\triangle ABC$ points $A, D, E,$ and $B$ lie that order on side $\overline{AB}$ with $AD = 4, DE = 16,$ and $EB = 8$. Points $A, F, G,$ and $C$ lie in that order on side $\overline{AC}$ with $AF = 13, FG = 52,$ and $GC = 26$. Let $M$ be the reflection of $D$ through $F$, and let $N$ be the reflection of $G$ through $E$. Quadrilateral $DEGF$ has area 288. Find the area of heptagon $AFNBCEM$. | 588 |

Table 7: Statistics and examples of the eight evaluation benchmarks utilized in the paper.

We present the statistics and examples of the eight evaluation benchmarks used in our work in Table 7. Among these, GSM8K [Cobbe et al., 2021] is the simplest, comprising grade school math word problems. The MATH-500 [Hendrycks et al., 2021], Gaokao-2023 [Zhang et al., 2023], Olympiad-Bench [He et al., 2024], and AMC23 [MAA, a] benchmarks consist of high school mathematics problems spanning a wide range of topics and difficulty levels, while Minerva Math [Lewkowycz et al., 2022] may also include problems from other subjects. The AIME [MAA, b] benchmark is a prestigious high school mathematics competition that requires deep mathematical insight and precise problem-solving skills. An overview of all benchmarks is provided as follows.

- **GSM8K**: A high-quality benchmark comprising 8,500 human-written grade school math word problems that require multi-step reasoning and basic arithmetic, each labeled with a natural language solution and verified answer. The 1,319-question test set emphasizes sequential reasoning and is primarily solvable by upper-grade elementary school students.

- **MATH-500**: A challenging benchmark of 500 high school competition-level problems spanning seven subjects, including Algebra, Geometry, Number Theory, and Precalculus. Each problem is presented in natural language with LaTeX-formatted notation, offering a strong measure of mathematical reasoning and generalization across diverse topics.

- **Minerva-Math**:A high-difficulty math problem dataset consisting of 272 challenging problems. Some problems are also relevant to scientific topics in other subjects, such as physics.

- **Olympiad-Bench**: An Olympiad-level English and Chinese multimodal scientific benchmark featuring 8,476 problems from mathematics and physics competitions. In this work, we use only the pure language problems described in English, totaling 675 problems.

- **Gaokao-2023**: A dataset consists of 385 mathematics problems from the 2023 Chinese higher education entrance examination, professionally translated into English.

- **AMC23**: The AMC dataset consists of all 83 problems from AMC12 2022 and AMC12 2023, extracted from the AoPS wiki page. We used a subset of this data containing 40 problems.

- **AIME24 & 25**: Each set comprises 30 problems from the 2024 and 2025 American Invitational Mathematics Examination (AIME), a prestigious high school mathematics competition for top-performing students, which are the most challenging benchmarks used in our study. Each problem is designed to require deep mathematical insight, multi-step reasoning, and problem-solving skills.

# K  Prompts

## K.1  Prompt for Category Labeling

Listing 1: The prompt for labeling the categories for mathematical problems, utilizing a few-shot strategy in which each category is represented by a labeled demonstration.

```
# CONTEXT #
I am a teacher, and I have some high-level mathematical problems.
I want to categorize the domain of these math problems.

# OBJECTIVE #
A. Provide a concise summary of the math problem, clearly identifying
the key concepts or techniques involved.
B. Assign the problem to one and only one specific mathematical domain
.
The following is the list of domains to choose from:
<math domains>
["Intermediate Algebra", "Geometry", "Precalculus", "Number Theory", "
Counting & Probability", "Algebra", "Prealgebra"]
</math domains>

# STYLE #
Data report.

# TONE #
Professional, scientific.

# AUDIENCE #
Students. Enable them to better understand the domain of the problems.

# RESPONSE: MARKDOWN REPORT #
## Summarization
[Summarize the math problem in a brief paragraph.]
## Math domains
[Select one domain from the list above that best fits the problem.]

# ATTENTION #
 - You must assign each problem to exactly one of the domains listed
 above.
 - If you are genuinely uncertain and none of the listed categories
 applies, you may use "Other", but this should be a last resort.
 - Be thoughtful and accurate in your classification. Default to the
 listed categories whenever possible.
 - Add "=== report over ===" at the end of the report.

<example math problem>
**Question**:
Let $ n(\ge2) $ be a positive integer. Find the minimum $ m $, so that
 there exists $x_{ij}(1\le i ,j\le n)$ satisfying:
(1)For every $1\le i ,j\le n, x_{ij}=max\{x_{i1},x_{i2},...,x_{ij}\} $
 or $ x_{ij}=max\{x_{1j},x_{2j},...,x_{ij}\}.$
(2)For every $1\le i \le n$, there are at most $m$ indices $k$ with
$x_{ik}=max\{x_{i1},x_{i2},...,x_{ik}\}.$
(3)For every $1\le j \le n$, there are at most $m$ indices $k$ with
$x_{kj}=max\{x_{1j},x_{2j},...,x_{kj}\}.$
</example math problem>

## Summarization
The problem involves an \( n \times n \) matrix where each element \(
x_{ij} \) is constrained by the maximum values in its respective row
or column. The goal is to determine the minimum possible value of \( m
 \) such that, for each row and column, the number of indices
attaining the maximum value is limited to at most \( m \). This
```

problem requires understanding matrix properties, maximum functions, and combinatorial constraints on structured numerical arrangements.

## Math domains
Algebra

=== report over ===

</example math problem>
**Question**:
In an acute scalene triangle $ABC$, points $D,E,F$ lie on sides $BC, CA, AB$, respectively, such that $AD \perp BC, BE \perp CA, CF \perp AB$. Altitudes $AD, BE, CF$ meet at orthocenter $H$. Points $P$ and $Q$ lie on segment $EF$ such that $AP \perp EF$ and $HQ \perp EF$. Lines $DP$ and $QH$ intersect at point $R$. Compute $HQ/HR$.
</example math problem>

## Summarization
The problem involves an acute scalene triangle with three perpendicular cevians intersecting at the orthocenter. Additional perpendicular constructions are made from specific points on segment \( EF \), leading to an intersection at point \( R \). The goal is to determine the ratio \( HQ/HR \), requiring knowledge of triangle geometry, perpendicularity, segment ratios, and properties of the orthocenter.

## Math domains
Geometry

=== report over ===

</example math problem>
**Question**:
Three cards are dealt at random from a standard deck of 52 cards. What is the probability that the first card is a 4, the second card is a $\clubsuit$, and the third card is a 2?
</example math problem>

## Summarization
This problem involves calculating the probability of a specific sequence of events when drawing three cards from a standard 52-card deck without replacement. It requires understanding conditional probability, the basic rules of counting, and how probabilities change as cards are removed from the deck.

## Math domains
Counting & Probability

=== report over ===

</example math problem>
**Question**:
Let $x$ and $y$ be real numbers such that $3x + 2y \le 7$ and $2x + 4y \le 8.$ Find the largest possible value of $x + y.$
</example math problem>

## Summarization
This problem involves optimizing a linear expression \( x + y \) subject to a system of linear inequalities. It requires understanding of linear programming concepts, such as identifying feasible regions, analyzing boundary points, and determining the maximum value of an objective function within that region.

## Math domains
Intermediate Algebra

=== report over ===

</example math problem>
**Question**:
Solve
\[\arccos 2x - \arccos x = \frac{\pi}{3}.\] Enter all the solutions,
separated by commas.
</example math problem>

## Summarization
This problem requires solving a trigonometric equation involving
inverse cosine functions. The equation relates two expressions with \(
 \arccos(2x) \) and \( \arccos(x) \), and asks for all real solutions
satisfying the given identity. It involves knowledge of inverse
trigonometric functions, their domains, and properties, as well as
algebraic manipulation.

## Math domains
Precalculus

=== report over ===

</example math problem>
**Question**:
What perfect-square integer is closest to 273?
</example math problem>

## Summarization
The problem asks for the perfect square integer closest to 273. This
involves understanding the distribution and properties of perfect
squares, and comparing them with a given integer. It relies on number-
theoretic reasoning related to squares of integers and their proximity
 to a target number.

## Math domains
Number Theory

=== report over ===

</example math problem>
Voldemort bought $6.\overline{6}$ ounces of ice cream at an ice cream
shop. Each ounce cost $\$0.60.$ How much money, in dollars, did he
have to pay?
</example math problem>

## Summarization
The problem involves multiplying a repeating decimal, \( 6.\overline
{6} \), by a fixed unit price, \$0.60, to find the total cost in
dollars. This requires converting a repeating decimal into a fraction
or using decimal multiplication, both of which are foundational
arithmetic skills.

## Math domains
Prealgebra

=== report over ===

<math problem>
{problem}
</math problem>

## K.2 Prompt for Concepts Extraction

Listing 2: Prompt template for extracting internal concepts from a mathematical question.

```
As an expert in educational assessment, analyze this problem:
<problem>
{problem}
</problem>

Break down and identify {num_concepts} foundational concepts being
tested. List these knowledge points that:
- Are core curriculum concepts typically taught in standard courses,
- Are precise and measurable (not vague like "understanding math"),
- Are essential building blocks needed to solve this problem,
- Represent fundamental principles rather than problem-specific
techniques.

Think through your analysis step by step, then format your response as
 a Python code snippet containing a list of {num_concepts} strings,
where each string clearly describes one fundamental knowledge point.
```

## K.3 Prompt for Problem Synthesis

Listing 3: Prompt template for synthesizing math problems from specified concepts, difficulty levels, and pre-defined mathematical categories. Following [Zhao et al., 2025b], the difficulty levels are consistently set to the competition level to prevent the generation of overly simple questions.

```
### Given a set of foundational mathematical concepts, a mathematical
domain, and a specified difficulty level, generate a well-constructed
question that meaningfully integrates multiple listed concepts and
reflects the stated level of complexity.

### Foundational Concepts:
{concepts}

### Target Difficulty Level:
{level}

### Mathematical Domain:
{domain}

### Instructions:
1. Begin by outlining which concepts you will combine and how you plan
 to structure the question.
2. Ensure that the question is coherent, relevant, and appropriately
challenging for the specified level.
3. The question must be a single standalone problem, not split into
multiple sub-questions.
4. Do not generate proof-based, multiple-choice, or true/false
questions.
5. The answer to the question should be expressible using numbers and
mathematical symbols.
6. Provide a final version of the question that is polished and ready
for use.

### Output Format:
- First, provide your brief outline and planning for the question
design.
- Then, present only the final version of the question in the
following format:

'''
[Your developed question here]
```

```
, , ,

Do not include any placeholder, explanatory text, hints, or solutions
to the question in the output block
```

## K.4   Prompt for Quality Evaluation

Listing 4: The quality evaluation prompt utilized to filter out low-quality math problems. Following prior work [Zhao et al., 2025b], we assess synthetic problems based on five criteria: **format, factual accuracy, difficulty alignment, concept coverage, and solvability**. Each problem is then assigned one of three quality levels: **'bad', 'acceptable', or 'perfect'**.

```
As a critical expert in educational problem design, evaluate the
following problem components:

=== GIVEN MATERIALS ===
1. Problem & Design Rationale:
{rationale_and_problem}
(The rationale describes the author's thinking process and
justification in designing this problem)

2. Foundational Concepts:
{concepts}

3. Target Difficulty Level:
{level}

=== EVALUATION CRITERIA ===
Rate each criterion as: [Perfect | Acceptable | Bad]
1. FORMAT
- Verify correct implementation of markup tags:
<!- BEGIN RATIONALE -> [design thinking process] <!- END RATIONALE ->
<!- BEGIN PROBLEM -> [problem] <!- END PROBLEM ->

2. FACTUAL ACCURACY
- Check for any incorrect or misleading information in both problem
and rationale
- Verify mathematical, scientific, or logical consistency

3. DIFFICULTY ALIGNMENT
- Assess if problem complexity matches the specified difficulty level
- Evaluate if cognitive demands align with target level

4. CONCEPT COVERAGE
- Evaluate how well the problem incorporates the given foundational
concepts
- Check for missing concept applications

5. SOLVABILITY
- Verify if the problem has at least one valid solution
- Check if all necessary information for solving is provided

=== RESPONSE FORMAT ===
For each criterion, provide:
1. Rating: [Perfect | Acceptable | Bad]
2. Justification: Clear explanation for the rating

=== FINAL VERDICT ===
After providing all criterion evaluations, conclude your response with
:
'Final Judgement: [verdict]'
where verdict must be one of:
```

- 'perfect' (if both FACTUAL ACCURACY and SOLVABILITY are Perfect, at least two other
criteria are Perfect, and no Bad ratings)
- 'acceptable' (if no Bad ratings and doesn't qualify for perfect)
- 'bad' (if ANY Bad ratings)

Note: The 'Final Judgement: [verdict]' line must be the final line of your response.

