# OpenReview forum: "SwS: Self-aware Weakness-driven Problem Synthesis in Reinforcement Learning for LLM Reasoning"
_NeurIPS.cc/2025/Conference — NeurIPS 2025 poster_

### Official Review · Reviewer_faYg · 2025-07-01

**Clarity:** 3
**Significance:** 3
**Originality:** 3
**Rating:** 4
**Confidence:** 3

**Summary:**

The proposed Self-aware Weakness-driven Problem Synthesis (SwS) framework demonstrates innovation and strength. By leveraging the model's self-identified weaknesses and generating synthetic problems for training augmentation, it enhances the model's reasoning capabilities. Compared to existing problem synthesis strategies for LLM reasoning, the SwS targets the model’s capabilities and self-identified weaknesses, enabling improvement in RL training.

**Questions:**

1. Could the weakness-driven augmentation strategy (Section 4.3) lead to over-specialization by heavily sampling problems conceptually aligned with past failures, potentially causing the model to overfit to specific problem archetypes and hindering broader generalization?

2. Does the weakness criterion defined in Equation (1) (failure rate > 50% and declining accuracy trend) fail to capture critical scenarios? For instance, if the accuracy of a problem drops significantly from 80% to 30%, wouldn't this be excluded from the weakness set due to the initial high accuracy? Recommend adding parameter analysis experiments.

3. In the case presented in Figure 7, does the reorganization of "properties of equilateral triangles" and "points and planes in 3D space" to generate the simple problem stray too far from the core of the original problem? Could this lead to synthetic problems diverging from the model's true weaknesses? How can we ensure that invalid problems are avoided during the generation process?

4. In the geometric synthesis (Figure 7), the reorganization of "equilateral triangle properties" and "3D point-plane relations" from the original problem leads to a loss of the core equidistant condition in the simple problem and reduces spatial relationships to 2D in the medium problem, which seems to deviate from the original reasoning structure. Can the current reorganization mechanism fully capture the mathematical essence of the original problem, and how can invalid problems be avoided in the process?

5. The experiment only categorizes problems by accuracy (simple/medium/hard) but does not address their true cognitive difficulty in human evaluation. Could the low accuracy (6.25%) on "hard" problems be due to design flaws rather than inherent difficulty? It is recommended to include more examples for illustration.

**Ethical Concerns:**

["NO or VERY MINOR ethics concerns only"]

**Final Justification:**

Thanks to the authors for the rebuttal. The responses are beneficial to facilitating understanding of this paper. However, the authors are still suggested to incorporate the new discussions/experiments/clarifications mentioned in the rebuttal to the paper.

Therefore, I maintain my borderline accept rating.

**Limitations:**

Several aspects require further clarification. Please reply to my concerns raised in the ``Questions'' section.

**Quality:**

3

**Strengths And Weaknesses:**

- The SwS framework tackles model reasoning weaknesses by using self-identified gaps and targeted training to generate synthetic problems.

- The framework is tested across various mathematical reasoning benchmarks, demonstrating performance gains across different model sizes.

Weaknesses can be found in the question section. Some aspects require further discussion and clarification.

---

> ### Author Rebuttal · Authors · 2025-07-31
>
> We sincerely appreciate Reviewer faYg for recognizing our method as innovative and our experiments and analysis as comprehensive. Thanks for your time and thoughtful review, and we provide the following clarifications to address your concerns.
>
> ## 1. Response to overfitting to past failures (Q1）:
>
> Clarifications ✏️: We believe that **the weakness-driven augmentation is unlikely to cause overfitting** to past failures and forgetting. **Since the augmented training set includes the original dataset, such as MATH-12k for smaller models and DAPO + LightR1 for larger models, the model continues to be exposed to the broad distribution of their problems.** If the model forgets how to solve certain problems from the original dataset, during the subsequent augmented RL phase, it will still be required to solve those types of problems, leading to recovery and improvement in those areas. Moreover, if the model fails to learn certain problems from the original set during both the initial and augmented RL phases, it is likely due to the inherent difficulty of those problems rather than overfitting on weakness-driven augmented problems.
>
> Clarifications ✏️: The experiments in Section 3.3 and Figure 4 also show that **SwS-augmented training enables the model to solve more problems across all categories of the original dataset compared to training on the original dataset alone.** Notably, the model achieves 20% and 5% higher solve rates on previously unsolved problems in its weakest areas—Intermediate Algebra and Geometry, respectively. Together with the performance improvements on evaluation benchmarks in Table 1, these results demonstrate that the weakness-driven augmentation strategy in SwS promotes better generalization rather than overfitting to past failure cases.
>
> ## 2. Response to weakness criterion (Q2):
>
> Experiments 📊: We sincerely thank the reviewer for pointing out the potentially overlooked cases in weakness identification. To examine the mentioned scenario and evaluate the robustness of our failure definition in Equation 1, we conduct a statistical analysis of problems where accuracy drops from over 80% to below 30%, based on Qwen2.5-7B training on the MATH-12k dataset. **We find that only 0.5% of the problems in MATH-12k exhibit such performance drop.** Empirically, these drops are largely due to random fluctuations during training, and performance often recovers in subsequent epochs. Among them, **56.1% of the problems reach a minimum accuracy of at least 50% during augmented RL training epochs, and 84.8% achieve an average accuracy above 50% during those epochs.**
>
> Clarifications ✏️: **These statistics suggest that, for such edge cases not captured by our weakness criteria, the RLVR process during augmented training can effectively improve performance without requiring additional data augmentation.** While treating these cases as weaknesses might further enhance learning, it could also lead to concept-ineffectiveness in problem synthesis, which reduces the sampling probability of concepts associated with more severe failure cases. To ensure more effective targeting of true weaknesses, we therefore choose not to include such cases as weaknesses.
>
>
> ## 3. Response to potential divergence between synthetic and failure problems (Q3):
>
> Clarifications ✏️: We believe that the reorganization of the "properties of equilateral triangles" and "points and planes in 3D space" is necessary, as the original problem requires knowledge and skills related to the properties of equilateral triangles and dihedral angles. The problems we constructed from sampled concepts—-such as the “Simple”, "Medium" and “Hard” examples shown in Figure 7—can help us better understand the underlying fundamental concepts. This case study demonstrates that **it is possible to generate geometry-related questions of varying difficulty based on specific concepts.** Furthermore, by combining resampled concepts drawn from the model’s weaknesses, we can generate a large number of such questions. **Statistically, since the sampled concepts for problem synthesis originate from the base model's weaknesses, training on the synthesized problems comprehensively addresses the weaknesses exposed in the original training set.**
>
> Clarifications ✏️: Admittedly, some synthetic questions may be unsolvable or ineffective for RLVR training. To mitigate this issue in augmentation, we **first employ an powerful reasoning model to label the answers using self-consistency and filter out questions with low consistency, which are likely either too difficult or unanswerable. Additionally, we use the initially trained policy model to assess whether the difficulty of the synthetic problems is appropriate for RLVR training, ensuring that all remaining synthetic problems provide effective learning signals during augmented training.** You may refer to lines 448–470 and Table 4 in our appendix for our whole data workflow, where we describe how we filtered out basic formatting issues as well as problems that were too simple or too difficult:
> ```
> (1) Removing multiple-choice, multi-part, or proof-required problems;
> (2) Discarding problems evaluated as low quality;
> (3) Filtering out problems where the answer generation model yields inconsistent answers, specifically when the most frequent answer among all generations appears less than 50%; and
> (4) Removing problems whose difficulty levels are unsuitable for the current model in RL training.
> ```
>
> ## 4. Response to concept capture and potential invalid questions (Q4):
>
> Clarifications ✏️: Actually, it is challenging to strictly guarantee that the knowledge structure of synthetic problems is fully aligned with that of the failure cases, **as the concepts extracted from failure cases are shuffled within their respective categories and resampled for problem synthesis.** However, we perform large-scale sampling to ensure that the number of synthetic problems is sufficient to comprehensively cover the identified weaknesses. For example, **as shown in Figure 8, starting from a relatively small set of around 1,900 seed problems, we were able to generate a large number of synthetic problems—up to 1,000,000 in total—resulting in over 40,000 problems after filtering.** Some of these may exhibit high similarity to the failure cases. For categories with a higher proportion of failure cases, we allocated more sampling capacity to increase the likelihood that the synthetic problems share similar knowledge structures with the original ones.
>
> ## 5. Response to human evaluation and potential design flaws (Q5):
>
> Clarifications ✏️: Regarding the alignment between human-evaluated and model sampling-based difficulty, **since the SwS pipeline is designed to mitigate the weaknesses of a specific model, the assessment of problem difficulty should be based on the model’s current reasoning capabilities.** Given that each model has different strengths and weaknesses in reasoning, a problem may vary in difficulty across models—just as humans have varying proficiencies with certain problems—making human-evaluated difficulty inherently subjective and difficult to standardize. Consequently, **relying on other models or human evaluations, rather than the model itself, could result in inconsistency between problem augmentation and the goal of mitigating model weaknesses.**
>
> Futher more, from the perspective of the GRPO used in our paper, its main training strategy is to reinforce the model’s correct reasoning paths while suppressing incorrect ones. The difficulty filtering stage aims to ensure that problem difficulty matches the current model’s capability, allowing the model to generate both correct and incorrect answers, thus providing effective learning signals. Human-evaluated difficulty may not align with this objective, as the model might produce zero accurate rollouts on problems deemed difficult by humans.
>
> Experiments 📊: To better illustrate the discrepancy between the current model’s difficulty evaluation and external assessments, we conduct a simulated difficulty evaluation using the problems shown in Figure 7. Mimicking human difficulty evaluation, we prompt GPT-4.5 to assess these problems using the following criteria and assign difficulty scores from 1 to 10.
> ```
> 1. **Conceptual Complexity**: What level of math is needed to understand the problem (e.g., arithmetic, algebra, geometry, combinatorics, calculus)?
> 2. **Number of Steps / Depth of Reasoning**: How many steps does it take to solve the problem? Are they straightforward or require abstraction?
> 3. **Cognitive Load / Trickiness**: Are there traps, misconceptions, or misleading structures? Is it easy to make a mistake?
> 4. **Calculation Complexity**: Does it require long or error-prone computations?
> ```
> The GPT-4.5 evaluated difficulty scores for these problems from Simple to Unsolvable are [5, 6, 8, 5]. The first three evaluations align with sampling-based difficulty, whereas the evaluation of the Unsolvable problem does not. **If augmentation relies on such evaluation scores, the Unsolvable problem—where the model fails to sample an accurate answer—would provide no useful training signal during GRPO training, leading to data inefficiency.** Therefore, using the model’s own sampling-based difficulty evaluation would be more targeted on its weaknesses and data-efficient.
>
> For better clarity, we will include additional case studies across all categories of synthetic problems (Simple/Medium/Hard) in appendix.

---

> > ### Comment · Reviewer_faYg · 2025-08-04
> > **I maintain my rating.**
> >
> > Thanks to the authors for the rebuttal. The responses are beneficial to facilitating understanding of this paper. However, the authors are still suggested to incorporate the new discussions/experiments/clarifications mentioned in the rebuttal to the paper.
> >
> > Therefore, I maintain my borderline accept rating.

---

> > > ### Author Response · Authors · 2025-08-05
> > > **Thanks for Reviewer faYg's feedback**
> > >
> > > We sincerely thank you for considering our efforts during the rebuttal period. We will incorporate all the experiments and summarized clarifications from the rebuttal into the updated version of the paper. Thank you again for your thorough review and valuable feedback.

---

> ### Author Response · Authors · 2025-08-04
> **Kindly Request Feedback from Reviewer faYg**
>
> Dear Reviewer faYg,
>
> We sincerely appreciate the time and effort you devoted to reviewing our manuscript. In response to your thoughtful feedback, we have submitted a rebuttal with extensive experimental results addressing your concerns, which includes the following key points:
>
> + **Potential overfitting to past failures**: We clarified that SwS augmentation is not prone to overfitting to past failures, as the original training set is incorporated into the augmented training for ensuring generalization. Experiments in Section 3.3 and Figure 4 also demonstrate that SwS promotes better generalization.
> + **Potertial missing failures in our weakness criterion**: We studied the issue you mentioned, where the accuracy of certain problems drops sharply from high to low, and found that such cases are rare (0.5%). In most instances, the model regains high accuracy on these problems during subsequent RL training. Therefore, these cases do not represent true weaknesses and may also lead to concept-ineffectiveness in problem synthesis.
> + **Potertial divergence from failures and avoiding invalid problems**: We clarified that SwS augmentation statistically covers identified failure cases by extensively sampling concepts from them. To avoid invalid problems, we apply comprehensive rule-based filtering, self-consistency answer-labeling using a reasoning model, and difficulty filtering based on the initialized policy.
> + **Regarding to human-evaluation difficulty**: We clarified that since the SwS pipeline is designed to mitigate the weaknesses of a specific model, the assessment of problem difficulty should be based on the model’s current reasoning capabilities. We also provided an example using GPT-4.5 to simulate human evaluation, illustrating the limitations of human-based difficulty assessment for augmentation purposes.
>
> We believe these clarifications and supplementary experiments may help address your concerns. We welcome any further feedback or discussions and would appreciate it if you could consider revising the score positively in light of these illustrations. Thank you once again for your valuable feedback and for taking our response into consideration during the rebuttal process.
>
> Sincerely,
>
> Authors of Paper #20069

---

### Official Review · Reviewer_1yRo · 2025-07-02

**Clarity:** 3
**Significance:** 3
**Originality:** 4
**Rating:** 4
**Confidence:** 3

**Summary:**

This paper introduces the Self-aware Weakness-driven problem Synthesis (SwS) framework, a novel approach for improving the mathematical reasoning capabilities of Large Language Models (LLMs) through Reinforcement Learning with Verifiable Rewards (RLVR). The core contribution is a method that moves beyond generic data augmentation by first identifying a model's specific "weaknesses"—problems it consistently fails to solve during an initial RL training phase. The framework then extracts the core mathematical concepts from these failure cases, recombines them to synthesize a new, targeted set of problems, and uses this augmented dataset for subsequent training. The authors demonstrate that this weakness-driven approach allows the model to focus on and overcome its deficiencies more efficiently. Through extensive experiments on models ranging from 3B to 32B parameters across eight reasoning benchmarks, the SwS framework shows significant performance gains, improving average scores by up to 10.0% for 7B models and 7.7% for 32B models over strong baselines. The paper also explores extensions of the framework to weak-to-strong generalization and self-evolving paradigms, showcasing its robustness and adaptability.

**Questions:**

This is generally strong work, in my opinion. However, I have a few points that need clarification from the authors, and I would appreciate the inclusion of some ablation experimental results.


1. The criteria for identifying a "failed problem" in Equation 1 are defined by two specific thresholds: an accuracy below 50% and a negative slope in performance over time. How sensitive is the framework's overall performance to these specific values? Have you experimented with alternative definitions, such as a static low-accuracy threshold or different regression criteria, and how did they compare?

2. The SwS pipeline appears to be computationally intensive, involving an initial RL run, concept extraction, problem generation with a 70B model, quality verification, and answer generation with a 32B model. Could you provide a more detailed breakdown of the computational cost (e.g., GPU hours) for each stage of the pipeline? This would be very helpful for reproduction and follow-up work.

3. The weak-to-strong experiment is a promising direction. However, the "weak teacher" (Qwen2.5-Math-7B-Instruct) is still a somewhat powerful model. How does the framework's difficulty filtering and overall performance degrade as the capability gap between the teacher and the student widens significantly? Is there a point where an unreliable teacher leads to the filtering of nearly all synthetic problems, or worse, the introduction of noise that harms the student model?

**Ethical Concerns:**

["NO or VERY MINOR ethics concerns only"]

**Final Justification:**

I am keeping my initial score, same as Reviewer faYg.

**Limitations:**

It seems that there is a lack of discussion regarding the paper's limitations. A "Limitation" section would be appreciated.

**Paper Formatting Concerns:**

The appendix appears to lack some details regarding the experimental setup, and I believe additional experiments should be included for a more comprehensive analysis.

**Quality:**

3

**Strengths And Weaknesses:**

## Strengths

- Clarity: The paper is well-written, clearly structured, and easy to follow. Figures and diagrams are highly effective at illustrating the complex methodology.

- Performance: The reported improvements are substantial and consistent across different model scales, providing strong support for the paper's central claims.

- Analysis: The paper includes valuable analyses on weakness mitigation, the impact of question difficulty, and extensions to weak-to-strong learning, adding considerable depth to the work.

## Weaknesses

- Complexity and Cost: The SwS pipeline is computationally expensive and complex, requiring multiple powerful LLMs for its various stages. This presents a potential hurdle for reproducibility and practical deployment for teams with fewer resources.

---

> ### Author Rebuttal · Authors · 2025-07-31
>
> We sincerely thank Reviewer 1yRo for considering our paper well-written, with solid experiments and comprehensive analysis. We greatly appreciate your time and thoughtful review, and we address your concerns as follows, particularly regarding the computational cost.
>
> ## 1. Response for the potential complexity of the SwS framework (W1):
>
> Clarifications ✏️: In the SwS pipeline, we leverage SOTA open-source instruction and reasoning models to generate high-quality problems along with corresponding answers. To reduce computational overhead, we propose two alternative strategies for augmentation based on model weaknesses under RLVR. First, **since all models in SwS are modular and interchangeable, in resource-constrained settings, users can utilize smaller yet powerful models equipped with thinking capabilities**, such as Qwen3-4B [1], which can still generate high-quality questions using our prompts. Below, we provide an example of using Qwen3-4B to synthesize problems from specific concepts and categories. **Secondly, users can adopt our proposed weakness-driven selection strategy in Section 4.3** to augment the training set, thereby avoiding the heavy costs associated with full data synthesis pipeline.
>
> ```
> ### Input
> Concepts:
> ['Knowledge of strategies for solving systems of equations, including substitution and elimination methods',
>  'Direction vectors of lines',
>  'Understanding of the definitions and properties of trigonometric functions, including sine, cosine, and tangent',
>  'Understanding of ratios and proportions, including the ability to simplify and manipulate ratios',
>  'Knowledge of basic trigonometric identities, including sum-to-product formulas and Pythagorean identities']
>
> Category: Precalculus
>
> ### Output:
> <think>
> Okay, let's tackle this problem. The user wants a question that integrates several foundational concepts from precalculus at the AMC12 level. The concepts listed are solving systems of equations (substitution and elimination), direction vectors of lines, trigonometric functions (sine, cosine, tangent), ratios and proportions, and trigonometric identities like sum-to-product and Pythagorean.
>
> ......
>
> This problem requires solving the system of equations (from the intersection and perpendicular condition), using trigonometric identities, and applying the tangent addition formula. It integrates all the required concepts.
> </think>
>
> '''text
> Two lines intersect at a point (x, y) where x/y = 2. One line has direction vector (cos θ, sin θ) and passes through the origin, while the other has direction vector (cos φ, sin φ) and passes through (1, 0). The lines are perpendicular. Find the value of tan(θ + φ).
> '''
> ```
>
>
> ## 2. Reponses for the criteria of failed problems (Q1):
>
> Clarifications ✏️: We sincerely thank you for the suggestion to more comprehensive analysis of Eq.1. For the accuracy threshold, intuitively, **increasing the threshold in Eq.1 may introduce additional unnecessary problems,** potentially reducing the efficiency of leveraging concepts truly related to model weaknesses in data synthesis. Conversely, **lowering the threshold could lead to missed model weaknesses and a reduction in the diversity of synthesized problems.**
>
> Experiments 📊: To better understand the sensitivity of the failure criteria, we further analyze **the negative slope constraint.** We study the weakness identification experiments of Qwen2.5-7B model on MATH-12k, focusing on problems with best accuracy ≤ 50% and a positive slope (in contrast to negative slope used in Eq.1). **These problems represent one-third of all problems with best accuracy ≤ 50% (660 out of 1997). We find that over half (350 of 660) start with 0% accuracy in the first epoch, but their performance improves during early RL training, suggesting that additional augmentation may not be necessary for them.** Tracking the accuray of these problems with best accuracy ≤ 50% under baseline RL, **we observe that only 4.24% of problems with a positive slope are never correctly answered during later RL training (steps 200–800), whereas 32% of problems with a negative slope consistently fail. This indicates that problems with a positive initial slope can be solved through extended training, while those with a negative slope more accurately represent true model weaknesses that naive RL cannot overcome.** Therefore, we incorporate the negative slope constraint into the failure definition in Eq. 1. However, due to computational resource limits (as discussed in the next response) and time constraints, we regret that we could not comprehensively evaluate alternative failure definitions during rebuttal.
>
>
> ## 3. Reponses for the detailed computational cost (Q2):
>
> Clarifications ✏️: For the specific GPU hours at each stage, we use the Qwen2.5 7B experiment as an example and report the GPU hours for each SwS stage in the table below. All time measurements are based on NVIDIA A100 40G GPUs. Notably, **the total time spent on all problem synthesis stages (8,292.6 GPU hours) is actually less than that required by the final augmented training via RL (9,543 GPU hours). This comparison highlights the rationale for allocating computational resources to data augmentation prior to RL.** Within the problem synthesis pipeline, the most time-consuming component is Answer Generation, as it requires a powerful reasoning model to ensure answer correctness. In contrast, other stages mainly involve shorter inference, thus require less inference time.
>
> Clarifications ✏️: Importantly, unlike prior works that rely on proprietary APIs [2, 3], **all models used in our experiments are open-source, making the pipeline more cost-effective.** Futhermore, **users can freely substitute them with smaller models or follow [4] to train lightweight models for problem generation.** Alternatively, to avoid computation-intensive problem generation, users may adopt our proposed weakness-driven selection strategy in Section 4.3 to augment the training set.
>
> ### GPU Hours for SwS Experiments with Qwen2.5-7B
> |  | Weakness Identification | Concepts Extraction | Problem Generation | Quality Verification | Answer Generation | Difficulty Filtering | Augmented Training |
> | --- | --- | --- | --- | --- | --- | --- | --- |
> | GPU Hours (h) | 2,075 | 1.6 | 1,940 | 975  | 4,608 | 768 | 9,543 |
> | Data Quantity  | 12k | 1339 | 1000k | 842k | 302k | 253k | 52k |
> | Data / Hour  | - | 836.9 | 515.5 | 863.6 | 65.5 | 329.4 | - |
>
>
> ## 4. Reponses regarding the weak teacher model in Section 4.1 (Q3):
>
> Experiments 📊: To analyze the case where the student model differs more significantly from the teacher model, **we use Qwen2.5-3B-Instruct to guide the same student model as in Section 4.1.** The answer labeling statistics are presented in the following table, where the first four rows correspond to the Qwen2.5-7B-Math-Instruct experiments in Table 5, Appendix E. **SC** refers to the teacher model’s self-consistency evaluation, while **Stu-Con** indicates whether the student model produces a sufficient proportion of answers consistent with the teacher model in difficulty filtering. Similarly, by applying self-consistency and filtering out low-consistency answers, the accuracy of the weak teacher-labeled answers increases from 62.4% to 83.6%.
>
> However, it is still 10.8% lower than the results of Qwen2.5-7B-Math-Instruct, introducing a 10.8% labeling error rate (i.e., noise). Meanwhile, the number of problems with consistency above 50% drops from 444 (Qwen2.5-7B-Math-Instruct) to 354, indicating that using a weaker model may lose more difficult yet key problems.
>
> Experiments 📊: In addition, we observe that **using a weaker model can lead to an imbalanced distribution in the labeled dataset.** As shown in the last row of the table below, which reports the proportion of problems retained after teacher self-consistency and student difficulty filtering, the 3B teacher model performs relatively well on Algebra, retaining 91% of the problems. However, due to its limited capabilities, it fails to retain many problems in categories such as Precalculus (50%) and Geometry (58%). **One potential strategy is to utilize multiple weaker teacher models—each with different strengths but generally smaller and less capable than the student model—to enable weak-to-strong generalization,** similar to the strategy in [5]. That is, a weaker teacher model should possess some strengths over the student model; if so, it can effectively guide the student from certain perspectives. In contrast, a teacher model that is uniformly weaker across all dimensions may be ineffective.
>
> Clarifications ✏️: In addition to using an external teacher model, **our experiments in Section 4.2 (Self-Evolving) demonstrate that the base model itself can serve as the answer-labeling teacher through self-consistency.** Along with Section 4.1, these results show that the teacher model used for answer labeling does not necessarily need to outperform the student model, highlighting the robustness of the SwS pipeline.
>
> ### Experiments using Qwen2.5-3B-Instruct as Teacher Model
>
> | Setting | Prealgebra | Intermediate Algebra | Algebra | Precalculus | Number Theory | Counting & Probability | Geometry | All |
> | --- | --- | --- | --- | --- | --- | --- | --- | --- |
> | Pass@1  | 73.2 | 35.1 | 85.5 | 48.2 | 69.4 | 55.3 | 51.2 | 62.4 |
> | SC | 79.3 | 53.6 | 91.9 | 51.8 | 87.1 | 71.1 | 61.0 | 73.2 |
> | SC>50% | 90.6 | 60.8 | 91.2 | 82.1 | 83.7 | 80.0 | 83.3 | 83.6 |
> | SC>50% & Stu-Con | 91.9 | 60.8 | 91.2 | 82.1 | 84.8 | 80.0 | 83.3 | 84.0 |
> | Retained Ratio  | 76% | 53% | 91% | 50% | 74% | 66% | 58% | 69.8% |
>
>
> ## References
>
> [1] Qwen3 technical report
>
> [2] Key-point-driven data synthesis with its enhancement on mathematical reasoning.
>
> [3] Common 7b language models already possess strong math capabilities.
>
> [4] Promptcot: Synthesizing olympiad-level problems for mathematical reasoning in large language models.
>
> [5] Language Model Preference Evaluation with Multiple Weak Evaluators

---

> ### Author Response · Authors · 2025-08-04
> **Kindly Request Feedback from Reviewer 1yRo**
>
> Dear Reviewer 1yRo,
>
> We sincerely appreciate the time and effort you devoted to reviewing our manuscript. In response to your thoughtful feedback, we have submitted a rebuttal with extensive experimental results addressing your concerns, which includes the following key points:
>
> + **Potential complexity of our framework**: Regarding the complexity of SwS, to reduce computational cost, we propose two alternative augmentation methods: first, using a lightweight instruction model with strong reasoning capabilities in the problem synthesis pipeline; second, adopting our weakness-driven selection strategy described in Section 4.3 of the paper.
> + **Analysis on the criteria of failed problems**: We have studied each component of the failure problem definition, with a focused analysis on the use and necessity of the slope constraint for better understanding their roles.
> + **Detailed computational cost**: We provide comprehensive details (GPU hours) of each stage in SwS of our experiments on Qwen2.5-7B. We find that the whole problem synthesis requires fewer GPU hours than augmented RL training, indicating that performing data augmentation before RL is both reasonable and cost-effective.
> + **Analysis on the weak teacher model in Section 4.1**: To analyze the impact of using a weaker teacher model with a larger gap relative to the student, we added a new experiment using the Qwen2.5-3B-Instruct model and compared its effectiveness to the Qwen2.5-7B-Math-Instruct model.
>
> We hope that these clarifications and additional experiments effectively address your concerns. We kindly invite any further feedback or questions, and respectfully hope you could consider positively re-evaluating the score in light of these updates. Thank you again for your valuable feedback and for considering our response during the rebuttal process.
>
> Sincerely,
>
> Authors of Paper #20069

---

> > ### Comment · Reviewer_1yRo · 2025-08-05
> >
> > Dear authors,
> >
> > Thank you very much for your response and strong feedback. I have no further questions. I will look into other reviewers' comments for the final rating.

---

> ### Author Response · Authors · 2025-08-05
> **Thanks for Reviewer 1yRo's review and feedback**
>
> Thank you very much for considering our efforts during the rebuttal period. We’re pleased if our responses have addressed some of your concerns. We also did our best to address the issues and clarify misunderstandings raised in other reviews, which all received positive feedback. Please feel free to check about them, and we welcome any additional questions from you. Finally, we sincerely appreciate your thoughtful review and valuable feedback for this work, and hope to hear good news from you~

---

### Official Review · Reviewer_iWhd · 2025-07-03

**Clarity:** 1
**Significance:** 2
**Originality:** 2
**Rating:** 4
**Confidence:** 4

**Summary:**

The paper proposes a method for improving LLMs on mathematical reasoning by synthesizing problems based on specific topics where they fail the most (their "weaknesses"). The method first runs RL training on an initial set of mathematical problems. Then, it computes a failure rate for each problem based on both the success rate on that problem being always low (< 0.5) and moreover decreasing during RL training (negative slope in success rate). The LLM then extracts topics from those hard problems, synthesizes new problems from those topics, and then is trained via RL on a combination of the original and synthetic problems. Experiments training Qwen models from 3B to 32B show improvements on a range of mathematical reasoning datasets.

**Questions:**

* How exactly do you compute the slope in Eq 1 (L106-107)? Have the authors tried using only the first component of accuracy < 50%? What fails if the slope is indeed important here?
* How do you compute Avg@32? Intuitively, Pass@32 should be generally higher than Pass@1, but this does not seem to be the case in AIME. L192 makes it sound like you're using Avg and Pass as synonyms, but this seems inconsistent with the results.
* Have the authors tried iterating the procedure, i.e., repeating SwS after one cycle of weakness detection, problem generation and training? If so, do you get further gains?
* Why are the other datasets besides AIME and GSM8k missing from Table 2? The reference to Appendix C in L242 is also incorrect - I assume the authors meant Appendix E, which says more about the generalization experiment.

**Ethical Concerns:**

["NO or VERY MINOR ethics concerns only"]

**Final Justification:**

The authors pointed to existing and new results that address some of my concerns regarding the limitations and use of a single training dataset. I think the paper provides extensive experiments and pushes on an interesting direction, so I'm slightly recommending its acceptance. That said, the most interesting results still consist of marginal improvements (e.g., on the weak-to-strong and self-training settings, where they do not assume a stronger teacher model). For that reason I am not assigning a higher score.

**Limitations:**

The paper doesn't mention limitations very explicitly - for instance, it doesn't mention that all experiments are done by training models of the same family.

**Quality:**

2

**Strengths And Weaknesses:**

The paper tackles an important problem of improving LLM reasoning abilities without the need to collect further human data. The idea of focusing on specific categories of model failures is interesting. The authors also took care of not generating unreasonable combinations of topics by leveraging co-occurrence statistics of topics in existing problems.

My main high-level concerns about the paper are (1) using a single training dataset, MATH, and (2) using of a single model family during training (Qwen models).

The training procedure seems to rely heavily on the initial training problems already being labeled with categories. The categories in MATH, like "Algebra", are also fairly broad. This seems to really limit the current method to the coverage of the initial training dataset. While the method synthesizes new problems, the authors also seem to focus on problems that the initial model already has *some* chance of solving (e.g., by the Difficulty Filtering, which ignores problems with a smaller than .25 success rate). Thus, it seems like the method can improve a model's chance of solving those problems where it is uncertain, but not teach it anything completely new, or really address its worst weaknesses.

Moreover, using a single model family for training limits the extent to which the paper can claim to address specific model weaknesses during training. For this, you would need to take different models with very different failure profiles, and show that SwS adapts what it does to those particular model failures. Otherwise, it might just be that SwS turns out to be an effective data synthesis procedure to address the needs of Qwen models, but it might be that (a) the same dataset would also similarly improve other models, like DeepSeek or LLaMA, which would weaken the claim that SwS is addressing model failures that are specific to Qwen, or that (b) SwS would fail to help other model families. Thus, the experiments can't yet support the main claims of the paper.

---

> ### Author Rebuttal · Authors · 2025-07-31
>
> We appreciate that Reviewer iWhd finds our topic interesting and experiments concrete. We find the concerns are mostly due to misunderstandings. We have addressed all concerns below and thank you very much for your time and careful review.
>
> ## 1. Response to single training dataset and model family (W1):
>
> Clarifications ✏️: Regarding using only MATH for training, **we have also used the DAPO [1] and LightR1 [2] datasets for larger models,** as detailed in L186–188. **The category labels in them were generated using the prompt in Appendix I.** Generally, we select training datasets based on models' reasoning capabilities. Specifically, we found that Qwen2.5-7B and smaller models show consistent improvements when trained on MATH-12k. However, the 14B and 32B models quickly saturate on MATH-12k in early training. Therefore, we chose more challenging datasets, including DAPO and LightR1, that better stimulate the reasoning abilities of stronger models.
>
> Clarifications ✏️: Regarding the concern about using only the Qwen model family, we primarily chose Qwen due to their strong reasoning capabilities compared to other open-source base models [3]. Some recent studies have discussed the failure cases when applying RLVR training to other base models. **In [4], the authors find that the LLaMA base model lacks significant reasoning patterns like verification and backtracking, thus their performance improvement from RLVR significantly lagged behind Qwen. Besides, recent mid-training work [5,6] also reports that LLaMA bases are ill-suited to zero-RL, performing far below the Qwen family, where the RLVR trainining fails to eliciting the LLaMA base model's reasoning capabilities over its few-shot settings [6].**
>
> Experiments 📊: To demonstrate the comprehensiveness of our SwS framework on models beyond Qwen family, we additionally conduct experiments using the LLaMA-3.1-8B-Instruct model. Specifically, we use MATH-12k for 200 steps RL in Weaknesses Identification, then apply the SwS pipeline to synthesize 18k problems for an additional 300 steps of augmented RL training, while continuing the baseline model for the same 300 steps. The following table shows that SwS augmentation leads to a 2.3% overall improvement.
>
> |  | GSM8k | MATH | Minerva | Olympiad | Gaokao | AMC23 | AIME24 | AIME25 | Avg |
> | --- | --- | --- | --- | --- | --- | --- | --- | --- | --- |
> | LLaMA-3.1-8B-IT | 85.6 | 48.2 | 24.6 | 18.8 | 39.7 | 22.5 | 6.7 | 3.3 | 31.1 |
> | Initial RL | 88.9 | 56.0 | 29.2 | 21.7 | 48.6 | 30.0 | 16.7 | 6.7 | 37.2 |
> | Baseline RL | 88.3 | 58.4 | 31.2 | 23.4 | 49.6 | 30.0 | 16.7 | 6.7 | 38.0 |
> | SwS-LLaMA | 90.5 | 60.2 | 33.5 | 25.8 | 49.1 | 40.0 | 16.7 | 6.7 | 40.3 |
>
> ## 2. Response to SwS augmentation could not learn new capabilities (W2):
>
> Clarifications ✏️: Regarding the concern about learning new capablities, we have discussed it with experiments in Section 3.3 (L212–L222, Figure 4). The y-axis of Figure 4 represents the percentage of problems in the original training set **where the model never generates an accurate solution throughout the entire training process, which reflect the base model’s worst weaknesses. The experimental results show that the SwS augmentation enables the model to solve certain problems that cannot be learned through training on the original set alone,** particularly in its weakest areas such as Intermediate Algebra and Geometry, where the SwS-augmented model achieves 20% and 5% relative higher solve rates, respectively. This demonstrates that our SwS augmentation is effective not only in improving performance on uncertain cases but also in eliciting the model’s deeper reasoning capabilities in its most challenging domains.
>
> ## 3. Response to the slope conputation and its necessity (Q1):
>
> Clarifications ✏️: We sincerely thank you for pointing out the clarification needed regarding the slope calculation in Eq.1. For computing the slope in Eq.1 (L106-107), we compute the slope using a **least-squares linear regression over the sequence of accuracy recorded across training epochs**, implemented using `np.linalg`. This metric reflects the model’s learning progress on a problem throughout the training process. We will include this clarification in Appendix B in the updated version.
>
> Experiments 📊: To assess the impact of the negative slope condition in Eq.1, we study on Qwen2.5-7B's weakness identification results using MATH-12k. **We examine problems with best accuracy ≤ 50% and a positive slope** (in contrast to negative slope in Eq.1). These problems account for one-third of all problems (660/1997) with best accuracy ≤ 50%. We find that over half (350/660) of these problems start with 0% accuracy in the first epoch. However, their performance improves during the initial RL stage, suggesting that learning such problems may not require additional augmentation. We further track the accuracy of these 1997 problems with best accuracy ≤ 50% in later stage of the baseline RL. **We find that only 4.24% of problems with a positive initial slope were never answered correctly by the model during later RL training (step 200 to 800), while 32% of problems with a negative initial slope were consistently failed.** This result indicates that problems with a best accuracy of ≤ 50% and a positive initial slope can be solved through extended training, **whereas those with a negative slope more accurately reflect the model's true weaknesses that naive RL can not learn.** Therefore, we set the negative slope constraint into the failure definition in Eq.1.
>
> ## 4. Responses to "Avg@k" and "Pass@k" metric in the paper (Q2):
>
> Revision 📄: We admit that the metric labeled as Pass@32 in Table 2 should actually be Avg@32, and we will correct this in the updated version. Regarding L192, when k=1, Avg@1 and Pass@1 are equivalent because only a single answer is considered. We will clarify our notation by consistently using Avg@k to avoid any potential confusion in the updated version.
>
> ## 5. Responses to repeating SwS after one cycle (Q3):
>
> Experiments 📊: Thank you for the valuable suggestion regarding the iterative SwS setting. This approach could significantly enhance the extensibility of SwS. To understand the potential improvements from iterating the SwS pipeline, **we select failure cases using Eq.1 during the first augmented RL stage and use them to synthesize 18k new problems for a second augmentation stage,** following the SwS pipeline. We run the second augmentation stage for 300 steps, while extend the first augmentation for the same steps. The comparison results are in the following table.
>
> Notably, the second SwS iteration yields a 1.5% absolute improvement in overall performance. Meanwhile, **we also observed that updating training sets during RL helps the model maintain an appropriate level of training entropy,** which is linked to sustainable exploration and potential gains in RLVR training [7]. Specifically, the training entropy dropped to 0.02 by the end of the first augmentation stage. However, during the second augmentation stage, the entropy initially increased from 0.02 to 0.7 within 200 steps before starting to decline. This observation not only shows strong potential that a new round of SwS augmentation can yield performance gains, but also offers insights into when such augmentation need to be updated and are most effective. We plan to further explore it in future work.
>
> |  | GSM8k | MATH | Minerva | Olympiad | Gaokao | AMC23 | AIME24 | AIME25 | Avg |
> | --- | --- | --- | --- | --- | --- | --- | --- | --- | --- |
> | Stage-1 Aug | 93.9 | 82.6 | 41.9 | 49.6 | 71.7 | 67.5 | 26.7 | 20.0 | 56.7 |
> | Stage-1 Aug-Extend | 94.1 | 83.4 | 42.3 | 52.4 | 72.2 | 70.0 | 26.7 | 20.0 | 57.6 |
> | Stage-2 Aug | 94.3 | 84.4 | 44.5 | 50.1 | 73.7 | 72.5 | 30.0 | 23.3 | 59.1 |
>
> ## 6. Response to benchmarks in Table 2 and appendix misreference (Q4):
>
> Revision 📄: We thank you for pointing out the misreference in L242 and apologize for the confusion. The correct reference should be Appendix E, and we will revise it in the updated version.
>
> Clarifications ✏️: For other benchmarks not included in Table 2, we did not intend to present them here. **Our intention in constructing this table was showing that a teacher model, though weaker in certain areas (lower AIME24), with strengths in specific domains (multiple MATH-500 categories), can effectively guide the student model to achieve overall better performance—both in AIME24 and multiple MATH-500 subcategories.** This illustrates that a teacher model does not need to be generally stronger than the student in SwS, demonstrating its robustness. For clarification, we present the full benchmark results of the student model, teacher model, and trained student model in the following table. **The results align with Table 2 in the paper, demonstrating that a teacher model with weaker performance on AMC and AIME can still effectively enhance the student model’s performance on such competition-level benchmarks.**
>
> |  | GSM8k | MATH | Minerva | Olympiad | Gaokao | AMC23 | AIME24 | AIME25 | Avg |
> | --- | --- | --- | --- | --- | --- | --- | --- | --- | --- |
> | Strong Student | 92.0 | 77.7 | 36.3 | 38.4 | 62.9 | 50.0 | 10.2 | 8.3 | 47.0 |
> | Weak Teacher | 93.3 | 80.6 | 36.8 | 36.6 | 64.9 | 45.0 | 7.2 | 6.2 | 46.3 |
> | Trained Student | 93.6 | 83.3 | 39.7 | 47.6 | 68.3 | 65.0 | 17.5 | 12.5 | 53.4 |
>
> ## References
>
> [1] Dapo: An open-source llm reinforcement learning system at scale.
>
> [2] Light-r1: Curriculum sft, dpo and rl for long cot from scratch and beyond.
>
> [3] SimpleRL-Zoo: Investigating and Taming Zero Reinforcement Learning for Open Base Models in the Wild.
>
> [4] Cognitive behaviors that enable self-improving reasoners, or, four habits of highly effective stars.
>
> [5] Behavior Injection: Preparing Language Models for Reinforcement Learning.
>
> [6] OctoThinker: Mid-training Incentivizes Reinforcement Learning Scaling.
>
> [7] The entropy mechanism of reinforcement learning for reasoning language models

---

> > ### Comment · Reviewer_iWhd · 2025-08-04
> >
> > Thank you for the clarifications and pointing to existing and new results that address some of my concerns regarding the limitations and use of a single training dataset. The results I find most interesting, such as in the self-evolving setting and the weak-to-strong experiments, are still relatively marginal. But that said, I think the paper provides enough experiments on several fronts and pushes on an interesting direction, so I revised my score to recommend acceptance.

---

> ### Author Response · Authors · 2025-08-04
> **Kindly Request Feedback from Reviewer iWhd**
>
> Dear Reviewer iWhd,
>
> We sincerely appreciate the time and effort you devoted to reviewing our manuscript. In response to your thoughtful feedback, we have submitted a rebuttal with extensive clarification and experimental results for addressing your concerns, which includes the following key points:
>
> + **Single training dataset and model family**: We have clarified that we also utilized DAPO and LightR1 training data to train stronger models, while we additionally conducted experiments on LLaMA-based models beyond the Qwen family for demonstrating the generalization of SwS.
> + **Potential for SwS learning new capabilities**: As shown in Section 3.3 (L212–L222, Figure 4), SwS augmentation enables the model to solve problems that cannot be learned from training on the original training set alone, demonstrating its generalizability and potential for introducing new capabilities.
> + **The slope calculation in Eq.1 and its necessity**: We have explained the slope calculation in Eq.1 and showed that problems with best accuracy ≤ 50% and a negative accuracy slope better reflect the true weaknesses of the base model.
> + **Experiments on SwS after one cycle**: Based on your insightful suggestions, we conducted experiments with a second round of SwS data augmentation and found it continued to improve performance. Additionally, it helped recover the declined training entropy and supported sustained RL learning.
> + **Remaining benchmarks in Table 2**: We present the performance of both teacher and student models on the remaining benchmarks and find that the results on other benchmarks are consistent with those in Table 2 of the paper.
>
> Meanwhile, we sincerely thank you for pointing out the typos and misreferences, which we will correct in the updated version. We hope these revisions address your concerns effectively and kindly ask you to reconsider the rating of our work in light of these updates. Thank you for considering our response during the rebuttal period.
>
>
> Sincerely,
>
> Authors of Paper #20069

---

> ### Author Response · Authors · 2025-08-05
> **Thanks for Considering our Clarifications and Updating Review**
>
> We sincerely thank you for your feedback and recommendation for acceptance. We are pleased that our efforts during the rebuttal period have effectively addressed some of your concerns. The motivation for extending SwS to the weak-to-strong and self-evolving settings lies in their potential to AGI, since there would be no stronger teacher models available for guiding the stongest model. A key limitation of these settings is the challenge of synthesizing difficult problems that the model itself cannot yet solve, and this may be mitigated through prompt-evolving strategies [8,9], which we plan to explore in future work. Once again, thank you for considering both our paper and our efforts during the rebuttal period.
>
> [8] Wizardmath: Empowering mathematical reasoning for large language models via reinforced evol-instruct
>
> [9] SAND-Math: Using LLMs to Generate Novel, Difficult and Useful Mathematics Questions and Answers

---

### Author Response · Authors · 2025-08-06
**Summary of the Rebuttal Period**

Thank you to all reviewers for their constructive feedback on our work. Below, we provide a brief overview of (1) the key contributions and strengths of our work, as recognized by the reviewers, and (2) the clarifications and additional updates in response to their comments.

## Summary of Strengths and Contributions
We propose a Self-aware Weakness-driven Problem Synthesis (SwS) framework that systematically identifies model reasoning weaknesses and leverages them for problem augmentation in reinforcement learning for LLM reasoning.
+ The SwS framework enhances LLM reasoning capabilities **without requiring human labeling** (Reviewers iWhd and faYg). Its extended **self-evolving and weak-to-strong settings** also push on an interesting direction. (Reviewers iWhd and 1yRo).
+ The paper includes **comprehensive analyses on each component** of the proposed SwS framework **as well as the extensions** (Reviewer 1yRo), while taking care of avoiding unreasonable synthetic problems (Reviewer iWhd).
+ We conduct **extensive experiments** on base LLMs of varying scales across multiple reasoning benchmarks (Reviewer 1yRo). Our approach consistently achieves significant and stable improvements (Reviewer faYg).

## Issues Addressed in the Rebuttal Period
In response to the reviewers' suggestions and concerns regarding our work, we have made efforts to address them in the following key aspects:
+ **Criteria for Failure Cases** (Reviewers 1yRo, iWhd, and faYg): We conducted comprehensive analysis of each constraint in Eq. 1, particularly emphasizing the necessity of the negative slope component. We also clarified that the potential for missing weaknesses is not a significant concern.
+ **Additional Base Models** (Reviewer iWhd): We further incorporated SwS with LLaMA-based models to enhance its comprehensiveness beyond the Qwen model family presented in our manuscript.
+ **Iterative SwS augmentation** (Reviewer iWhd): We conducted experiments with an additional round of SwS data augmentation and found it continued to improve performance. Additionally, it helped recover the training entropy and supported sustained RL learning.
+ **SwS Learn New Capabilities** (Reviewer iWhd): We clarified that SwS enables the model to solve problems that cannot be learned from training on the original training set alone (Section 3.3 L212–L222, Figure 4).
+ **Computation Efficiency** (Reviewer 1yRo): We provide two alternatives to reduce computational cost for SwS, including using a lightweight instruction model with strong reasoning capabilities, or adopting our proposed weakness-driven selection strategy. We also provide comprehensive details (GPU hours) of SwS, showing that the problem synthesis requires fewer GPU hours than augmented RL training.
+ **Human Difficulty Evaluation** (Reviewer faYg): We clarified that the SwS pipeline is designed to mitigate the weaknesses of a specific model, and using an example to illustrate that human difficulty evaluation may be inappropriate and lead to data inefficiency in SwS.

Thank you once again to all the reviewers for their feedback, which has helped us enhance our work to make it more solid and comprehensive.

---

### Decision · Program_Chairs · 2025-09-17

**Decision:**

Accept (poster)

**Comment:**

This paper introduces targeted data augmentation by identifying a model's specific weaknesses during RL and then synthesize new problems that address these deficiencies.

All reviewers found the paper's core contribution to be innovative and well-motivated, the experiments to be extensive, and the results to be significant across multiple benchmarks and model scales. The clarity of the presentation and the in-depth analyses were also highlighted as major strengths.

Reviewers' initial concerns, including the computational cost, the reliance on a single model family (Qwen), and the specific criteria used to define model "weaknesses," were generally addressed during author rebuttal. The authors presented new experimental results with LLaMA models, demonstrating the framework's generalizability. They also provided a detailed breakdown of the computational costs and offered analysis for their weakness identification method.

Overall, the paper proposes an effective solution for solving a highly relevant problem, making valuable contributions to the venue.